# How much correction is adequate? A Unified Bias-Aware Loss for Long-Tailed Semi-Supervised Learning

## Abstract

Long-tailed semi-supervised learning (LTSSL) suffers from class imbalance-induced biases in both training and inference. Existing debiasing methods typically rely on distribution priors, which fail to capture two critical dynamic factors: the pseudo-labeling-induced shifts in effective priors and the model's intrinsic evolving bias. To address this limitation, we propose Bias-Aware Loss (BiAL), a unified objective that replaces static distribution priors with the model's current bias. This straightforward substitution enables BiAL to generate plug-and-play bias-aware variants of cross-entropy/logit adjustment and contrastive heads, thereby unifying prior correction across diverse network architectures and training paradigms. Through theoretical analysis and empirical validation, we prove that our BiAL provides a singular, unified mechanism to align training with model's evolving state and achieves state-of-the-art performance on multiple datasets.

## 1 Introduction

Long-tailed datasets are ubiquitous in real-world recognition(Wei et al., 2024; Shi et al., 2023; Liu et al., 2019), a challenge further exacerbated in semi-supervised learning (SSL)(Zhang et al., 2021; Chen et al., 2023), where pseudo-labels amplify distributional imbalance(Wei et al., 2021b; Gan et al., 2024; Hong et al., 2021). Many existing researches, like logit adjustment (LA)(Menon et al., 2020), addresses such imbalance by incorporating distribution priors into training or inference.

However, these priors are static: they presuppose a fixed label distribution, which rarely holds in SSL(Rizve et al., 2021). This is because pseudo-labeling continuously shifts the effective class prior, and the model itself develops an evolving bias(Chen et al., 2022; Wang et al., 2019) that integrates signals from both labeled and unlabeled data. Consequently, correcting against a static prior may result in either under-correction or over-correction, as shown in Fig 2, thereby raising a critical question aligned with our core inquiry: ***How much correction is adequate?***

We contend that the "optimal" correction should precisely align with the bias inherently exhibited by the model's current training stage. Our key observation is that the model's class bias is measurable from no-information input: the resulting class probabilities form a stable estimate of its inductive bias, conditioned on the current training state, as shown in Fig 1. Based on this insight, we propose Bias-Aware Loss (BiAL), a unified objective that replaces static distribution priors with model's current bias and uses the debiased energy in place of raw logits throughout training and inference.

We substantiate BiAL through novel theoretical insights and comprehensive empirical evaluations. Theoretically, our analysis reframes theoretical motivation by showing why fixed-prior corrections become misspecified and how replacing them with the model's current bias yields a Bayes-aligned decision rule under balanced error with reduced stage-wise/cumulative regret. Empirically, across extensive experiments on diverse datasets, BiAL concurrently improves pseudo-label quality and test accuracy, consistently outperforming baseline methods.

In summary, our contributions are as follows:

1. A unified bias-aware objective: We introduce BiAL, which replaces static distribution priors with the model's current bias and uses the debiased energy across SSL losses, unifying correction via a single principle. BiAL also extends to supervised learning.

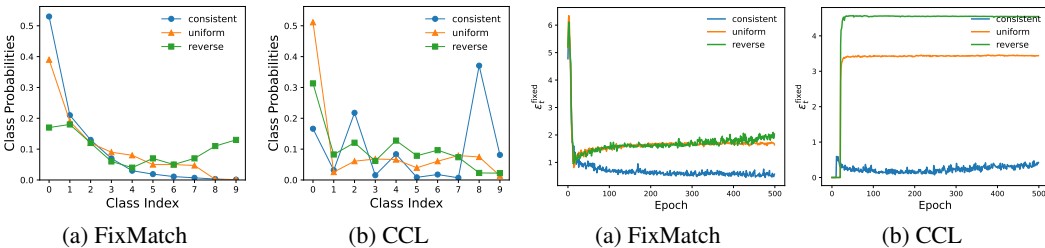

Figure 1: Class probabilities on an image without any patterns.

Figure 2: $\varepsilon_t^{\text{fixed}}$ 3.5 over epochs: worst-class multiplicative mismatch in log space; lower is better.

2. Theory for correctness and adaptivity: We establish Fisher consistency for balanced error with debiased energy, derive dynamic-regret advantages under prior drift, and provide a gradient-level rationale for reduced spurious correlations.

3. Simple, plug-and-play implementation with comprehensive validation: BiAL adds negligible overhead and no extra components, and achieves competitive results on long-tailed tasks across multiple datasets and distribution regimes.

## 2 RELATED WORK & PROBLEM SETUP

### 2.1 RELATED WORK

Modern SSL relies on high-confidence pseudo-labels with strong/weak consistency(Sohn et al., 2020; Berthelot et al., 2019; Fan et al., 2022), while under long-tailed distributions, pseudo-labels tend toward head classes and degrade tail recall. Some technologies mitigate bias by resampling or rebalancing(Cui et al., 2019; Lin et al., 2017) with an estimated unlabeled prior that is updated progressively; representative techniques include DARP(Kim et al., 2020; Kang et al., 2019), CReST+(Wei et al., 2021a), DASO(Oh et al., 2022), and ABC(Lee et al., 2021), which improve histogram balance yet still depend on an external or lagged proxy of true prior. Some other technologies apply bias-aware corrections at logit or energy level(Huang et al., 2016; 2019) and introduce expert heads; examples include LA-style compensation(Ren et al., 2020), ACR(Wei & Gan, 2023), CPE(Ma et al., 2024), and Meta-Experts(Hou & Jia, 2025), while post-hoc bias approaches such as LCGC(Xing et al., 2025) and CDMAD(Lee & Kim, 2024) estimate a classifier's bias(Wang et al., 2022a) from no-information inputs and use it for prediction correction or pseudo-label screening, typically without modifying the training loss. CCL(Zhou et al., 2024) offers a probabilistic view that unifies LA with class-balanced contrastive learning(Zhu et al., 2022; Cui et al., 2021) using reliable and smoothed pseudo-labels together with progressive estimation and alignment of the unlabeled label distribution. However, many LTSSL methods still rely on fixed or externally estimated priors, which can be brittle when the effective prior shifts over training as pseudo-labels are accepted.

### 2.2 PROBLEM SETUP

We consider $K$-way classification with a labeled set $\mathcal{D}_l = \{(x_i, y_i)\}_{i=1}^{n_l}$ and an unlabeled set $\mathcal{D}_u = \{u_j\}_{j=1}^{n_u}$, where $y_i \in [K]$. Let $f_\theta$ be the backbone and $z(x) \in \mathbb{R}^K$ the logits; posteriors are $p_\theta(y \mid x) = \text{softmax}(z(x))$. Denote the class priors of labeled and unlabeled data by $\pi^L, \pi^U \in \Delta^{K-1}$; the labeled per-class counts $\{N_c\}_{c=1}^K$ are sorted $N_1 \geq \cdots \geq N_K$ with imbalance ratio $\gamma_l = N_1/N_K$. Similarly, $\{M_c\}_{c=1}^K$ denote the (generally unknown) unlabeled per-class counts with imbalance ratio $\gamma_u = M_1/M_K$. Training minimizes a supervised loss on $\mathcal{D}_l$ and an SSL loss on $\mathcal{D}_u$:

$$\mathcal{L} = \mathbb{E}_{(x,y) \sim \mathcal{D}_l}\big[\ell_{\text{sup}}(z(x), y)\big] + \lambda \, \mathbb{E}_{u \sim \mathcal{D}_u}\big[\ell_{\text{ssl}}(z(u); \hat{q}(u))\big], \tag{1}$$

where $\hat{q}(u)$ is a (hard/soft) pseudo-label distribution produced by the current model. Since decisions are invariant to adding a constant to all logits ($z \leftarrow z + c\mathbf{1}$), any logit-based bias estimate is centered (mean-subtracted across classes) before use.

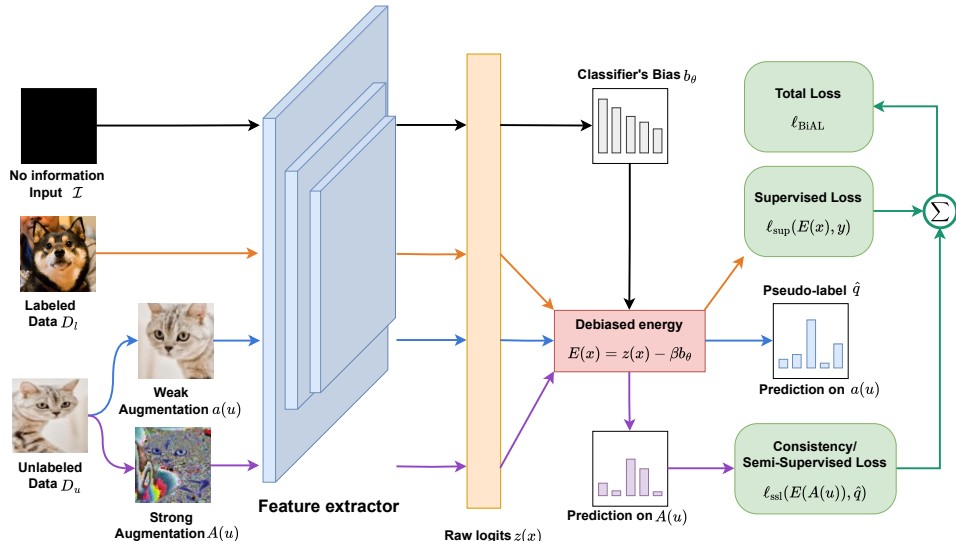

Figure 3: Training process of FixMatch using BiAL. We first use A lightweight bias-probing module to estimate the classifier's class-wise model-state bias $b_\theta$ from no-information images. Supervised and semi-supervised branches operate on a unified debiased energy $E = z - \beta\, b_\theta$. Furthermore, pseudo-labels are generated and filtered in this same debiased space to align with the evolving unlabeled prior; and the identical debiasing is applied at inference for train/test consistency.

## 3 Bias-Aware Loss: A Unified Loss For LTSSL

### 3.1 Model-induced bias: definition and estimation

The core of BiAL lies in tracking the model's current class bias $b_\theta \in \mathbb{R}^K$, which reflects the model's intrinsic inclination toward certain classes at the current training state. This bias is estimated exclusively from no-information inputs devoid of class-specific features to avoid conflating task-relevant signals with inductive bias.

**Definition of Bias.** Let $\mathcal{I}$ denote a distribution of no-information inputs. Consistent with empirical evidence that solid-color inputs yield optimal bias estimation(Xing et al., 2025), we use all-black images as $\mathcal{I}$. The bias $b_\theta$ is defined as:

$$\tilde{b}_{\theta,c} \;=\; \log\Big(\mathbb{E}_{I\sim\mathcal{I}}\, p_\theta(y{=}c \mid I)\Big), \qquad \bar{b}_\theta \;=\; \frac{1}{K}\sum_{c=1}^{K}\tilde{b}_{\theta,c}, \qquad b_\theta \;=\; \tilde{b}_\theta - \bar{b}_\theta \mathbf{1}. \tag{2}$$

Centering operation: Subtracting $\bar{b}_\theta \cdot \mathbf{1}$ eliminates arbitrary global shifts in logits. It is critical because softmax outputs are invariant to uniform logit shifts, ensuring $b_\theta$ only captures class-specific bias.

### 3.2 Debiased energy

To integrate the estimated bias into training, we define the debiased energy $E(x)$, a substitute for raw logits $z(x)$ that explicitly cancels the model's current class bias, which can be formulated as:

$$E(x) \;=\; z(x) \;-\; \beta\, b_\theta, \qquad \beta \geq 0. \tag{3}$$

where $\beta$ controls the strength of bias correction: a larger $\beta$ amplifies debiasing, while a smaller $\beta$ mitigates overcorrection. The detailed value analysis of $\beta$ can be found in the appendix.

Since $b_\theta$ captures the model's current class bias, subtracting $\beta b_\theta$ from raw logits $z(x)$ explicitly cancels this bias, ensuring the debiased energy $E(x)$ reflects task-relevant signals rather than the model's intrinsic inclination. This debiased energy is the "unifying bridge" of BiAL: it replaces raw logits in standard losses across SSL paradigms, ensuring consistent bias correction throughout training and inference.

### 3.3 PSEUDO-LABEL GENERATION

For SSL scenarios, pseudo-labels are generated using debiased energy instead of raw logits to align label proposal with BiAL's correction logic. The pseudo-labeling rule follows standard practice but operates on $E(x)$:

$$\hat{q}(u) = \begin{cases} \text{one-hot}\big(\arg\max_c s_c(u)\big), & \max_c s_c(u) \geq \tau_{\text{pl}}, \\ \text{ignore}, & \text{otherwise}, \end{cases} \tag{4}$$

where $s_c(u)$ denotes the score vector used to propose labels (in baseline methods $s = z$; in our method $s = E$). $\tau_{\text{pl}} \in [0, 1]$ is the confidence threshold(Wang et al., 2022b; Arazo et al., 2020).

### 3.4 UNIFIED LOSS FORMULATION

To unify the loss, we train by plugging $E(x)$ in equation 3 wherever a baseline would use raw logits $z(x)$:

$$\mathcal{L}_{\text{BiAL}}(\theta) = \underbrace{\mathbb{E}_{(x,y)\sim\mathcal{D}_l}\,\ell_{\text{sup}}\big(E(x), y\big)}_{\text{supervised}} + \lambda\,\underbrace{\mathbb{E}_{u\sim\mathcal{D}_u}\,\ell_{\text{ssl}}\big(E(u), \hat{q}(u; E)\big)}_{\text{semi-supervised}} \tag{5}$$

where $\ell_{\text{sup}}$ and $\ell_{\text{ssl}}$ are unchanged losses (CE/LDAM(Cao et al., 2019), FixMatch-style) but now consume $E$ instead of $z$. Pseudo-labels are generated from debiased energy: $\hat{q}(u; E)$ (top-1 or soft), not from $z$. Prior interpolation is optional, like EMA smoothness or to bridge static priors and model bias:

$$\tilde{b}_\theta = \lambda_B\,b_\theta + (1 - \lambda_B)\log\pi. \tag{6}$$

**Property 1 (reductions).**

(i) If $b_\theta = \log\pi + c\mathbf{1}$ and $\beta = \tau$, then $E(x) = z(x) - \tau\log\pi$ (logit adjustment).

(ii) If $\beta = 0$, BiAL recovers the baseline.

(iii) With $\lambda_B$ in equation 6, BiAL continuously interpolates between LA ($\lambda_B = 0$) and full bias-aware training ($\lambda_B = 1$).

**Property 2 (shift invariance).**

Since $b_\theta$ is centered and softmax is shift-invariant, adding any constant $c\mathbf{1}$ to $z(x)$ or $b_\theta$ leaves decisions and gradients unchanged. Therefore, BiAL does not introduce degenerate global shifts.

### 3.5 THEORETICAL MOTIVATIONS

**Pseudo-labels induce an evolving effective prior.**

In LTSSL, pseudo-labeling is the primary driver of dynamic shifts in the effective training prior, even if the latent distribution of unlabeled data remains fixed. To formalize this, we define two key components:

Acceptance-confusion matrix: Let $\hat{y}_t(u)$ denote the pseudo-label assigned to an unlabeled sample $u$ (weak augmentation) at training stage $t$, and $A_t(u) \in \{0, 1\}$ be an indicator for whether $\hat{y}_t(u)$ meets the confidence threshold (confidence $\geq \tau_{\text{pl}}$). The acceptance-confusion matrix $M_{y\to c}^t$ quantifies the probability that a sample with true class $y$ is assigned pseudo-label $c$ and accepted:

$$M_{y\to c}^t = \Pr\big(\hat{y}_t = c,\ A_t = 1 \,\big|\, Y = y\big) \tag{7}$$

Effective training prior: Let $\pi_t^U$ denote the latent (and typically unknown) class prior of unlabeled data at stage $t$. The effective prior $\tilde{\pi}_t^{PL}$ that the SSL loss actually "sees" is the distribution of accepted pseudo-labels, computed as:

$$\tilde{\pi}_t^{\text{PL}}(c) \;=\; \frac{\sum_y \pi_t^U(y)\, M_{y \to c}^t}{\sum_{y,c'} \pi_t^U(y)\, M_{y \to c'}^t}. \tag{8}$$

Critical to LTSSL, $\tilde{\pi}_t^{PL}$ drifts with training: $M_{y \to c}^t$ depends on the model's current prediction scores which evolve as training proceeds, so even if $\pi_t^U$ is fixed, $\tilde{\pi}_t^{PL}$ changes across stages. This dynamic shift breaks the core assumption of fixed-prior methods, rendering them ill-suited for LTSSL.

**Fixed-prior corrections are misspecified under drift.**

To quantify the excess error incurred by fixed-prior methods, we use the balanced error rate (BER), a critical metric for long-tailed tasks that balances error across classes. For stage $t$, the BER of a classifier $f$ is defined as:

$$\text{BER}_t(f) = \tfrac{1}{K} \sum_c \Pr\left(f(x) \neq c \mid Y = c\right) \tag{9}$$

Using Bayes theorem, the BER-optimal decision rule at stage $t$ is:

$$f_t^\star(x) = \arg\max_c \frac{\eta_c^t(x)}{\tilde{\pi}_t^{\text{PL}}(c)} = \arg\max_c \left(\log \eta_c^t(x) - \log \tilde{\pi}_t^{\text{PL}}(c)\right). \tag{10}$$

where $\eta_c^t(x) = \Pr\left(Y = c \mid X = x\right)$ is the posterior probability of class $c$ given input $x$.

Fixed-prior methods (LA) replace $\log \tilde{\pi}_t^{PL}$ with a static vector $g = \tau \log \pi^{\text{fixed}}$ (e.g., $\pi^{\text{fixed}} = \pi^L$, the labeled data prior). Their decision rule becomes:

$$f_t^{\text{fixed}}(x) = \arg\max_c \left(\log \eta_c^t(x) - \log \pi_c^{\text{fixed}}\right), \tag{11}$$

whose error depends on the log–prior mismatch $\varepsilon_t^{\text{fixed}} := \| \log \tilde{\pi}_t^{\text{PL}} - \log \pi^{\text{fixed}}\|_\infty$. A short distortion argument yields the bound: let $s_c(x) = \eta_c^t(x)/\tilde{\pi}_t^{\text{PL}}(c)$ and $\hat{s}_c(x) = \eta_c^t(x)/\pi_c^{\text{fixed}}$. Then $\hat{s}_c = r_c s_c$ with $r_c \in [e^{-\varepsilon_t^{\text{fixed}}}, e^{\varepsilon_t^{\text{fixed}}}]$. After normalizing $\bar{s}_c := s_c / \sum_j s_j$ and $\bar{\hat{s}}_c := \hat{s}_c / \sum_j \hat{s}_j$, we have $e^{-2\varepsilon_t^{\text{fixed}}} \bar{s}_c \leq \bar{\hat{s}}_c \leq e^{2\varepsilon_t^{\text{fixed}}} \bar{s}_c$, hence $\max_c \bar{\hat{s}}_c \geq e^{-2\varepsilon_t^{\text{fixed}}} \max_c \bar{s}_c$. Under the balanced–error criterion, $\text{BER}_t(f) = \mathbb{E}_X[1 - \max_c \bar{s}_c(X)]$ for the argmax rule, therefore:

$$\text{BER}_t\left(f_t^{\text{fixed}}\right) - \text{BER}_t\left(f_t^\star\right) \;\leq\; \frac{e}{K\, \pi_{\min}}\, \varepsilon_t^{\text{fixed}}. \tag{12}$$

Thus, any static prior incurs a stage-wise misspecification cost linear in its mismatch to the current effective prior; cumulating over stages yields excess error proportional to $\sum_t \varepsilon_t^{\text{fixed}}$.

**Bias-aware correction tracks drift and reduces excess BER.**

BiAL replaces $\log \pi^{\text{fixed}}$ by the model's current bias $b_t$, estimated from no-information inputs, centered, and EMA-smoothed, and uses debiased energies $E_t = z_t - \beta b_t$. Define $\varepsilon_t^{\text{BiAL}} := \|b_t - \log \tilde{\pi}_t^{\text{PL}}\|_\infty$. Repeating the same argument for $g = b_t$ yields the analogue of equation 3:

$$\text{BER}_t\left(f_t^{\text{BiAL}}\right) - \text{BER}_t\left(f_t^\star\right) \;\leq\; \frac{e}{K\, \pi_{\min}}\, \varepsilon_t^{\text{BiAL}}, \qquad f_t^{\text{BiAL}}(x) = \arg\max_c \left(\log \eta_c^t(x) - b_{t,c}\right). \tag{13}$$

Because the same biases that skew acceptance (and hence $M^t$) also appear when probing the model on no-information inputs, $b_t$ monotonically reflects $\log \tilde{\pi}_t^{\text{PL}}$ up to a class-independent shift (removed by centering); EMA further dampens estimation noise. Consequently $\varepsilon_t^{\text{BiAL}}$ is typically smaller than $\varepsilon_t^{\text{fixed}}$, giving lower per-stage and thus lower cumulative balanced error. The scalar $\beta$ simply reweights classes as $\tilde{\pi}^{\text{PL} - \beta}$; setting $\beta \approx 1$ aligns with BER, while a gentle ramp avoids early over-correction when $b_t$ is still noisy.

**Conclusion.**

In LTSSL, pseudo-labeling creates a drifting effective prior $\tilde{\pi}_t^{\text{PL}}$. Any fixed-prior correction is therefore inherently misspecified and provably incurs excess BER proportional to its log-prior mismatch. By substituting the model-induced bias $b_t$ for the prior at every stage, BiAL aligns the correction with the present state of the learner and reduces this mismatch term, providing a compact theoretical justification for the empirical gains we observe across consistent and inconsistent unlabeled regimes.

## 4 METHODOLOGY

This section details how we implement BiAL in practice. We first describe bias estimation and stabilization and then give drop-in recipes for semi-supervised (FixMatch/CCL) training.

### 4.1 ESTIMATING AND STABILIZING THE MODEL BIAS

**No-information baselines.** We probe the classifier on no-information inputs in normalized pixel space, for example, constant black images. Let the model output logits $z(I) \in \mathbb{R}^K$, we aggregate them with a numerically stable log-mean-exp and center the result to remove the softmax shift:

$$\tilde{b}_\theta = \log\Big(\frac{1}{|\mathcal{I}|}\sum_{I \in \mathcal{I}} \text{softmax}\big(z(I)\big)\Big) = \text{logsumexp}\big(\log\text{softmax}(z(I)); I\big) - \log|\mathcal{I}|,$$

$$b_\theta = \tilde{b}_\theta - \frac{1}{K}\big(\mathbf{1}^\top \tilde{b}_\theta\big)\mathbf{1}. \tag{14}$$

**Practical Estimation of $b_\theta$.** In practice, we estimate $\tilde{b}_\theta$ from a mini-batch of $|\mathcal{B}_I|$ no-information inputs per update and apply EMA smoothing:

$$b_\theta^{(t)} \leftarrow (1-\alpha)\, b_{theta}^{(t-1)} + \alpha\, \hat{b}_\theta^{(t)}, \tag{15}$$

where $\hat{b}_\theta^{(t)}$ is the current batch estimate and $\alpha \in (0, 1]$. We estimate the batch-level bias $\hat{b}_\theta^{(t)}$ from $|B_\mathcal{I}|$ at each update step, then update the global bias via EMA to suppress noise. We refresh $b_\theta$ every $E_{est}$ iterations or once per epoch instead of every step. This keeps computational overhead negligible.

**Warm-up and ramps.** Using a strong correction too early can cause oscillations. We therefore apply a ramp after a warm-up: $\beta_t$ for logit debiasing $z \leftarrow z - \beta_t b_\theta$. It follows a piecewise-linear schedule: zero during the first $E_{\text{warm}}$ epochs, then linearly ramp to target value over $E_{\text{ramp}}$ epochs:

$$r_t = \text{ramp}(t; E_{\text{warm}}, E_{\text{ramp}}) \in [0, 1], \quad \beta_t = \beta\, r_t. \tag{16}$$

This keeps early training close to the baseline and gradually introduces BiAL.

### 4.2 SSL FRAMEWORK ADAPTATION: FIXMATCH AND CCL

As shown in Fig 3, we implement BiAL by replacing the scores consumed by standard losses with debiased energies $E(x) = z(x) - \beta b_\theta$, where $b_\theta$ is the model's current class bias estimated from no-information inputs. The bias is refreshed every few iterations and gradually introduced by a warm-up followed by a linear ramp for the strengths $\beta_t$. This probe adds negligible overhead and leaves the architecture and optimizers unchanged.

Semi-supervised implementations keep the FixMatch or other model pipelines intact and substitute $E$ wherever scores are used. Pseudo-labels are proposed from the weak view using $p^E = \text{softmax}(E)$ and the same confidence threshold, and the strong view is trained against these labels with CE on $E$. For CCL prototype heads, we use bias-conditioned classwise temperatures, applied to both supervised and SSL branches. If a refinement module is present, it operates on $p^E$ to keep label generation aligned with the debiased energy used for learning.

---

**Algorithm 1** Semi-supervised BiAL (FixMatch/CCL)

---

**Inputs:** labeled set $\mathcal{D}_l$, unlabeled set $\mathcal{D}_u$; model $f_\theta$; epochs $T$; debias schedule $\{\beta_t\}$; SSL weight $\lambda$; confidence threshold $\tau_{\text{pl}}$; (optional) contrastive/prototype head.
**Output:** trained parameters $\theta$
 1: Initialize bias buffer $b \leftarrow \mathbf{0}$
 2: **for** $t = 1$ **to** $T$ **do**
 3:     **Estimate bias** $b_t$ from no-information inputs; center and EMA to update $b$
 4:     **Supervised branch:** for $(x, y) \sim \mathcal{D}_l$, compute $E(x) = z(x) - \beta_t b$; apply CE/LA or LDAM-on-$E$ to obtain $\mathcal{L}_{\text{sup}}$
 5:     **SSL proposal (weak view):** for $u \sim \mathcal{D}_u$, form $E(a(u))$ and $p^E(a(u)) = \text{softmax}(E(a(u)))$; if $\max_c p_c^E \geq \tau_{\text{pl}}$, set $\hat{y} = \arg\max p^E$
 6:     **SSL training (strong view):** compute $E(A(u))$; minimize $\mathcal{L}_{\text{ssl}} = \mathbf{1}[\hat{y}\text{ exists}] \cdot \text{CE}(\text{softmax}(E(A(u))), \hat{y})$
 7:     **Contrastive/Prototype:** if a contrastive-learning-style module is used, replace scores $s_c$ by $s_c - \beta_t b_c$ (or use bias-conditioned temperatures) and apply InfoNCE(Oord et al., 2018)/Proto-CE on $(x, A(u))$ to get $\mathcal{L}_{\text{ccl}}$
 8:     **Refinement:** if a Distribution-Alignment-style module is used, apply it *on* $p^E(a(u))$ to obtain refined targets
 9:     **Joint update:** minimize $\mathcal{L} = \mathcal{L}_{\text{sup}} + \lambda \mathcal{L}_{\text{ssl}} + \lambda_{\text{ccl}} \mathcal{L}_{\text{ccl}}$ (drop absent terms)
10: **end for**
11: **Return** $\theta$ *(test-time prediction can use $E = z - \beta b$)*

---

BiAL is engineered to be plug-and-play: a tiny probe to estimate $b_\theta$, two gentle ramps to avoid early instability, and a uniform $z \mapsto E = z - \beta b_\theta$ substitution across supervised and SSL heads, plus bias-aware margins/temperatures where applicable. These choices preserve the theoretical guarantees while making the method stable and easy to reproduce in modern pipelines.

## 5 EXPERIMENTS

We conducted comprehensive experiments to verify the effectiveness of BiAL on CIFAR10-LT(Krizhevsky et al., 2009), CIFAR100-LT(Krizhevsky et al., 2009), STL-10(Coates et al., 2011) and ImageNet-127 datasets(Fan et al., 2022). To simulate real-world unlabeled data, we tested our method on diverse distributions of unlabeled data.

### 5.1 SETUP

**Datasets.** We evaluate on CIFAR-10-LT, CIFAR-100-LT, STL10-LT and ImageNet-127. Following standard practice, let $\gamma$ be the imbalance ratio, and the labeled set $\mathcal{D}_l$ is made long-tailed with classes ordered by frequency. We conduct experiments under three regimes: **Consistent**, **Uniform**, **Reverse**. Due to limited space, the detailed experimental settings are deferred to the appendix.

### 5.2 RESULTS ON CIFAR10/100-LT

**Consistent distribution ($\gamma_l = \gamma_u$).**

In CIFAR-10/100-LT, replacing fixed-prior training with BiAL yields uniform gains for SSL. BiAL-FixMatch improves over the baseline, while BiAL-CCL achieves the strongest overall results and the clearest debiasing effect. This proves that BiAL makes models surpass strong LA-style baselines (CPE, Meta-Experts) under consistent settings.

**Inconsistent distributrion (uniform / reverse).**

When the unlabeled data depart from the labeled distribution, BiAL explicitly tracks model bias rather than assuming a static prior, making it robust to prior mismatch. BiAL-FixMatch consistently improves over the baseline under both uniform and reverse regimes; while BiAL-CCL achieves the highest performance, outperforming LA-based methods such as CPE that rely on prior anchoring.

Table 1: Test accuracy in consistent setting on CIFAR10-LT and CIFAR100-LT datasets. The underline is the best result for a single branch, and the best results are in **bold**.

| | CIFAR10-LT | | | | CIFAR100-LT | | | |
| | $\gamma_l = \gamma_u = 100$ | | $\gamma_l = \gamma_u = 150$ | | $\gamma_l = \gamma_u = 10$ | | $\gamma_l = \gamma_u = 20$ | |
| Algorithm | $N_1 = 500$ $M_1 = 4000$ | $N_1 = 1500$ $M_1 = 3000$ | $N_1 = 500$ $M_1 = 4000$ | $N_1 = 1500$ $M_1 = 3000$ | $N_1 = 50$ $M_1 = 400$ | $N_1 = 150$ $M_1 = 300$ | $N_1 = 50$ $M_1 = 400$ | $N_1 = 150$ $M_1 = 300$ |
|---|---|---|---|---|---|---|---|---|
| Supervised | 47.3±0.95 | 61.9±0.41 | 44.2±0.33 | 58.2±0.29 | 29.6±0.57 | 46.9±0.22 | 25.1±1.14 | 41.2±0.15 |
| w/ LA(Menon et al., 2020) | 53.3±0.44 | 70.6±0.21 | 49.5±0.40 | 67.1±0.78 | 30.2±0.44 | 48.7±0.89 | 26.5±1.31 | 44.1±0.42 |
| FixMatch(Sohn et al., 2020) | 67.8±1.13 | 77.5±1.32 | 62.9±0.36 | 72.4±1.03 | 45.2±0.55 | 56.5±0.06 | 40.0±0.96 | 50.7±0.25 |
| w/ DARP(Kim et al., 2020) | 74.5±0.78 | 77.8±0.63 | 67.2±0.32 | 73.6±0.73 | 49.4±0.20 | 58.1±0.44 | 43.4±0.87 | 52.2±0.66 |
| w/ CReST+(Wei et al., 2021a) | 76.3±0.86 | 78.1±0.42 | 67.5±0.45 | 73.7±0.34 | 44.5±0.94 | 57.4±0.18 | 40.1±1.28 | 52.1±0.21 |
| w/ DASOOh et al. (2022) | 76.0±0.37 | 79.1±0.75 | 70.1±1.81 | 75.1±0.77 | 49.8±0.24 | 59.2±0.35 | 43.6±0.09 | 52.9±0.42 |
| FixMatch + LA(Menon et al., 2020) | 75.3±2.45 | 82.0±0.36 | 67.0±2.49 | 78.0±0.91 | 47.3±0.42 | 58.6±0.36 | 41.4±0.93 | 53.4±0.32 |
| w/ DARPKim et al. (2020) | 76.6±0.92 | 80.8±0.62 | 68.2±0.94 | 76.7±1.13 | 50.5±0.78 | 59.9±0.32 | 44.4±0.65 | 53.8±0.43 |
| w/ CReST+(Wei et al., 2021a) | 76.7±1.13 | 81.1±0.57 | 70.9±1.18 | 77.9±0.71 | 44.0±0.21 | 57.1±0.55 | 40.6±0.55 | 52.3±0.20 |
| w/ DASO(Oh et al., 2022) | 77.9±0.88 | 82.5±0.08 | 70.1±1.68 | 79.0±2.23 | 50.7±0.51 | 60.6±0.71 | 44.1±0.61 | 55.1±0.72 |
| FixMatch + ABC(Lee et al., 2021) | 78.9±0.82 | 83.8±0.36 | 66.5±0.78 | 80.1±0.45 | 47.5±0.18 | 59.1±0.21 | 41.6±0.83 | 53.7±0.55 |
| w/ DASO(Oh et al., 2022) | 80.1±1.16 | 83.4±0.31 | 70.6±0.80 | 80.4±0.56 | 50.2±0.62 | 60.0±0.32 | 44.5±0.25 | 55.3±0.53 |
| FixMatch + CDMAD(Lee & Kim, 2024) | 80.3±0.21 | 83.6±0.46 | 73.3±0.63 | 80.5±0.76 | 50.6±0.44 | 60.3±0.32 | 44.7±0.14 | 54.3±0.44 |
| FixMatch + LCGC(Xing et al., 2025) | 81.2±0.73 | 83.9±0.36 | 74.3±1.92 | 80.8±0.32 | 50.9±0.45 | 60.2±0.57 | 44.6±0.81 | 55.3±0.48 |
| FixMatch + BiAL | 81.3±0.61 | 83.9±0.73 | 75.5±0.95 | 81.0±0.54 | 51.0±0.32 | 60.9±0.69 | 44.8±0.92 | 55.2±0.11 |
| FixMatch + ACR(Wei & Gan, 2023) | 81.6±0.19 | 84.1±0.39 | 77.0±1.19 | 80.9±0.22 | 51.1±0.32 | 61.0±0.41 | 44.3±0.21 | 55.2±0.28 |
| FixMatch + CPE(Ma et al., 2024) | 80.7±0.96 | 84.4±0.29 | 76.8±0.53 | 82.3±0.34 | 50.3±0.34 | 59.8±0.16 | 43.8±0.28 | 55.6±0.15 |
| FixMatch + Meta-Experts(Hou & Jia, 2025) | 81.7±0.39 | 84.6±0.19 | 77.2±0.58 | 82.5±0.40 | 50.9±0.41 | 60.3±0.29 | 44.2±0.29 | 55.9±0.83 |
| FixMatch + CCL(Zhou et al., 2024) | 84.5±0.38 | 85.5±0.35 | 81.5±0.99 | 84.0±0.21 | 53.5±0.49 | 63.5±0.39 | 46.8±0.45 | 57.5±0.16 |
| w/ BiAL-CCL | **85.0±0.39** | **86.5±0.98** | **81.9±0.65** | **84.5±0.20** | **53.8±0.57** | **63.9±0.43** | **47.1±0.22** | **57.9±0.21** |

Table 2: Test accuracy under inconsistent setting ($\gamma_l \neq \gamma_u$) on CIFAR10-LT and CIFAR100-LT datasets. $\gamma_l = 100$ for CIFAR10-LT, and $\gamma_l = 10$ for CIFAR100-LT dataset.

| | CIFAR10-LT ($\gamma_l \neq \gamma_u$) | | | | CIFAR100-LT ($\gamma_u =$ N/A) | | | |
| | $\gamma_u = 1$ (uniform) | | $\gamma_u = 1/100$ (reversed) | | $\gamma_u = 1$ (uniform) | | $\gamma_u = 1/10$ (reversed) | |
| Algorithm | $N_1 = 500$ $M_1 = 4000$ | $N_1 = 1500$ $M_1 = 3000$ | $N_1 = 500$ $M_1 = 4000$ | $N_1 = 1500$ $M_1 = 3000$ | $N_1 = 50$ $M = 400$ | $N_1 = 400$ $M = 300$ | $N_1 = 50$ $M = 400$ | $N_1 = 400$ $M = 300$ |
|---|---|---|---|---|---|---|---|---|
| FixMatch | 73.0±3.81 | 81.5±1.15 | 62.5±0.94 | 71.8±1.70 | 45.5±0.71 | 58.1±0.72 | 44.2±0.43 | 57.3±0.19 |
| w/ DARP | 82.5±0.75 | 84.6±0.34 | 70.1±0.22 | 80.0±0.93 | 43.5±0.95 | 55.9±0.32 | 36.9±0.48 | 51.8±0.92 |
| w/ CReST | 83.2±1.67 | 87.1±0.28 | 70.7±2.02 | 80.8±0.39 | 43.5±0.30 | 59.2±0.25 | 39.0±1.11 | 56.4±0.62 |
| w/ CReST+ | 82.2±1.53 | 86.4±0.42 | 62.9±1.39 | 72.9±2.00 | 43.6±1.60 | 58.7±0.16 | 39.1±0.77 | 56.4±0.78 |
| w/ DASO | 86.6±0.84 | 88.8±0.59 | 71.0±0.95 | 80.3±0.65 | 53.9±0.66 | 61.8±0.98 | 51.0±0.19 | 60.0±0.31 |
| w/ CDMAD | 87.5±0.46 | 90.3±0.27 | 79.3±0.78 | 84.2±0.31 | 54.8±0.19 | 63.3±0.24 | 51.2±0.30 | 61.7±0.54 |
| w/ LCGC | 88.1±0.72 | 91.0±0.37 | 80.1±0.66 | 85.1±0.66 | 55.7±0.93 | 64.1±0.44 | 51.5±0.62 | 62.5±0.85 |
| BiAL-FixMatch | 88.3±0.25 | 91.3±0.49 | 80.7±0.38 | 85.9±0.51 | 56.1±0.81 | 64.5±0.93 | 51.5±0.94 | 62.7±0.32 |
| FixMatch + ACR | 92.1±0.18 | 93.5±0.11 | 85.0±0.99 | 89.5±0.17 | 57.9±0.56 | 65.8±0.91 | 51.7±0.22 | 63.3±0.17 |
| FixMatch + CPE | 92.3±0.17 | 93.3±0.21 | 84.8±0.88 | 89.3±0.11 | 58.1±0.47 | 66.3±0.13 | 52.4±0.20 | 63.5±0.34 |
| FixMatch + CCL | 93.1±0.21 | 93.9±0.12 | 85.0±0.70 | 89.8±0.31 | 59.8±0.28 | 67.9±0.70 | 54.4±0.14 | 64.7±0.22 |
| BiAL-CCL | **93.4±0.25** | **94.1±0.34** | **86.0±0.67** | **90.2±0.24** | **60.5±0.58** | **68.2±0.99** | **55.0±0.41** | **65.2±0.28** |

## 5.3 Results on STL10-LT and ImageNet-127

We evaluate BiAL on STL10-LT where the label distribution of the unlabeled data is inherently inaccessible. As summarized in Table3, BiAL consistently improves its base learners under both $\gamma_l$ settings, yielding higher test accuracy; the CCL branch benefits further from training on debiased energies. Overall, the qualitative trend on STL10-LT indicates that our method improves pseudo-label quality and downstream generalization through debiasing at the score level.

We follow standard practice for ImageNet-127(Fan et al., 2022) and evaluate in the consistent setting used by prior work ($\gamma_l = \gamma_u \approx 286$). In this regime, representation quality and a balanced classifier are both critical. Integrating BiAL into the CCL branch preserves the original architecture and losses while aligning training and inference through debiased energies. As shown in Table 4, the BiAL-CCL is competitive on both resolutions.

## 5.4 Ablation Study

We ablate where BiAL is applied in semi-supervised training for both backbones by toggling it on the labeled branch, the unlabeled branch and test inference, which yields five concise regimes. All comparisons keep settings identical to the corresponding baselines, so improvements isolate the placement of BiAL rather than tuning.

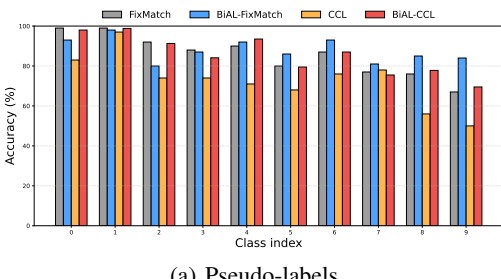

(a) Pseudo-labels

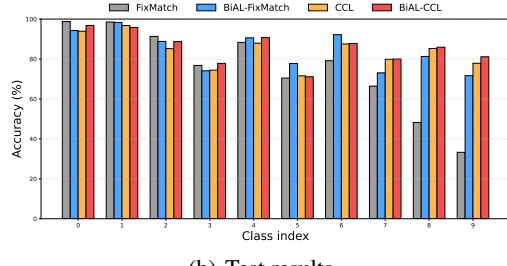

(b) Test results

Figure 4: The per-class accuracy of pseudo-labels and test results for FixMatch, BiAL-FixMatch, CCL and BiAL-CCL on CIFAR10-LT dataset in consistent settings ($\gamma_l = \gamma_u = 100$).

Table 3: Test accuracy on STL10-LT datasets.

| | STL10-LT ($\gamma_u$ = N/A) | | | |
|---|---|---|---|---|
| | $\gamma_l = 10$ | | $\gamma_l = 20$ | |
| Algorithm | $N_1 = 150$ $M = 100k$ | $N_1 = 450$ $M = 100k$ | $N_1 = 150$ $M = 100k$ | $N_1 = 450$ $M = 100k$ |
| FixMatch | 56.1±2.32 | 72.4±0.71 | 47.6±4.87 | 64.0±2.27 |
| w/ DARP | 66.9±1.66 | 75.6±0.45 | 59.9±2.17 | 72.3±0.60 |
| w/ CReST | 61.7±2.51 | 71.6±1.17 | 57.1±3.67 | 68.6±0.88 |
| w/ CReST+ | 61.2±1.27 | 71.5±0.96 | 56.0±3.19 | 68.5±1.88 |
| w/ DASO | 70.0±1.19 | 78.4±0.80 | 65.7±1.78 | 75.3±0.44 |
| w/ CDMAD | 72.5±0.39 | 79.9±0.23 | 66.3±0.57 | 75.2±0.40 |
| w/ LCGC | 72.8±0.61 | 80.1±0.42 | 66.5±0.83 | 76.6±0.34 |
| BiAL-FixMatch | 73.1±0.59 | 80.4±0.53 | 66.8±0.36 | 77.0±0.79 |
| FixMatch + ACR | 77.1±0.24 | 83.0±0.32 | 75.1±0.70 | 81.5±0.25 |
| FixMatch + CPE | 73.1±0.47 | 83.3±0.14 | 69.6±0.20 | 81.7±0.34 |
| FixMatch + CCL | 79.1±0.43 | 84.8±0.15 | 77.1±0.33 | 83.1±0.18 |
| BiAL-CCL | **79.8±0.51** | **85.2±0.21** | **77.6±0.45** | **83.7±0.28** |

Table 4: Test accuracy on ImageNet-127. The best results are in bold.

| ImageNet-127 ($\gamma_l = \gamma_u$) | | |
|---|---|---|
| Algorithm | $32 \times 32$ | $64 \times 64$ |
| FixMatch | 29.7 | 42.3 |
| w/ DARP | 30.5 | 42.5 |
| w/ DARP+cRT | 39.7 | 51.0 |
| w/ CReST+ | 32.5 | 44.7 |
| w/ CReST++LA | 40.9 | 55.9 |
| w/ CoSSL | 43.7 | 53.9 |
| w/ TRAS | 46.2 | 54.1 |
| w/ LCGC | 49.0 | 60.1 |
| w/ ACR | 57.2 | 63.6 |
| w/ CCL | 61.5 | 67.8 |
| BiAL-CCL | **62.0** | **68.2** |

Table 5: Ablation of BiAL placement on CIFAR-10-LT ($\gamma_\ell = \gamma_u = 100$) and CIFAR-100-LT ($\gamma_\ell = \gamma_u = 10$). Whenever the labeled or unlabeled branch uses BiAL during training, test-time debiasing is also enabled to align decision rules.

| | BiAL-FixMatch | | BiAL-CCL | |
|---|---|---|---|---|
| Condition | CIFAR-10 | CIFAR-100 | CIFAR-10 | CIFAR-100 |
| Baseline (no BiAL at train nor test) | 61.9 | 41.2 | 85.5 | 57.3 |
| Labeled-only + Test | 76.8 | 49.1 | 86.2 | 57.5 |
| Unlabeled-only + Test | 83.7 | 49.4 | 85.8 | 57.7 |
| Test-only (post-hoc on baseline model) | 81.7 | 52.3 | 85.7 | 57.5 |
| Full BiAL (Labeled + Unlabeled + Test) | **83.9** | **55.2** | **86.5** | **57.9** |

## 6 CONCLUSION

We present BiAL, a unified bias-aware objective that replaces static distribution priors with the current bias of the model and applies the resulting debiased energy consistently across semi-supervised learning. By aligning correction with the learner's state, BiAL provides theoretical guarantees: it achieves Fisher consistency with respect to balanced error and reduces dynamic regret under prior drift. Experiments on different datasets demonstrate that BiAL improves pseudo-label quality and test accuracy, and it integrates as a plug-and-play component without additional model complexity. These findings offer a simple and general recipe for robust long-tailed semi-supervised learning and motivate future work on stronger bias estimation and deeper integration with representation learning.

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

APPENDIX

# A  UNIFIED LOSS INSTANTIATIONS

Below we instantiate equation 5 for common families. The recipe is always: replace $z$ by $E$, and, where applicable, derive class-wise hyperparameters from $b_\theta$ to obtain a bias-aware version.

## A.1  BIAS-AWARE CROSS-ENTROPY / LOGIT ADJUSTMENT (BiAL-CE/LA)

**Supervised.**

$$\ell_{\text{sup}}^{\text{BiAL-CE}}(E, y) \;=\; -\log \frac{\exp\left(E_y\right)}{\sum_c \exp\left(E_c\right)} \;=\; -\log \frac{\exp\left(z_y - \beta b_{\theta,y}\right)}{\sum_c \exp\left(z_c - \beta b_{\theta,c}\right)}. \tag{17}$$

This is exactly CE on debiased logits. Standard LA corresponds to choosing $b_\theta = \log\pi$ and $\beta = \tau$.

**Semi-supervised (FixMatch-style).**

Let $a(u), A(u)$ be weak/strong augmentations and $p^E = \text{softmax}(E)$. With confidence $\tau_{\text{pl}}$:

$$\hat{y} = \arg\max_c p_c^E(a(u)), \quad \mathbb{1}_u = \mathbb{1}\left[\max_c p_c^E(a(u)) \geq \tau_{\text{pl}}\right], \tag{18}$$

$$\ell_{\text{ssl}}^{\text{BiAL-CE}} = \mathbb{1}_u \, \text{CE}\left(p^E(A(u)), \, \text{one} - \text{hot}(\hat{y})\right). \tag{19}$$

Using $E$ for both proposal and training aligns pseudo-labeling with the debiasing principle.

## A.2  BIAS-AWARE LDAM (BiAL-LDAM)

LDAM applies a class-dependent margin $m_y$ to the target logit. Classic LDAM uses static counts $n_y$ via $m_y \propto n_y^{-1/4}$. We define a soft, bias-induced effective count $\tilde{n}_c \propto \exp(b_{\theta,c})$, renormalized so $\sum_c \tilde{n}_c = \sum_c n_c$, or simply absorb the scale into $m_0$, and set

$$m_c(b_\theta) \;=\; m_0\, \tilde{n}_c^{-1/4} \;=\; m_0 \exp\left(-\tfrac{1}{4}\, b_{\theta,c}\right) \quad \text{(with clamping } m_{\min} \leq m_c \leq m_{\max}). \tag{20}$$

Then the BiAL-LDAM supervised loss is

$$\ell_{\text{sup}}^{\text{BiAL-LDAM}}(E, y) = -\log \frac{\exp(E_y - m_y(b_\theta))}{\exp(E_y - m_y(b_\theta)) + \sum_{c \neq y} \exp(E_c)}. \tag{21}$$

- If $b_\theta$ tracks majority inclination (larger $b_{\theta,c}$,c for majority), equation 20 reduces their margin and increases tail margins.
- When $b_\theta = \log n + c\mathbf{1}$, equation 21 reduces to classic LDAM (up to scale $m_0$).

## A.3  BIAS-AWARE CONTRASTIVE/CCL HEADS (BiAL-CCL)

Two plug-in variants are common; both debias either the logits or the temperature using $b_\theta$.

**(A) Debiased logits for class-prototype heads.**

Let similarity logits be $s_c(x) = \gamma \cos(h(x), \mu_c)$ (projection $h$, prototypes $\mu_c$, scale $\gamma$). Replace

$$s_c^{\text{BiAL}}(x) \;=\; s_c(x) \;-\; \beta\, b_{\theta,c}, \qquad \ell_{\text{sup}}^{\text{BiAL-Proto}} = -\log \frac{\exp(s_y^{\text{BiAL}}(x))}{\sum_c \exp(s_c^{\text{BiAL}}(x))}. \tag{22}$$

This is the prototype analogue of equation 17.

**(B) Class-wise temperatures.**

Keep logits $s_c(x)$ but use $\tau_c$ as a monotone function of bias, e.g.

$$\tau_c \;=\; \tau_0 \,\exp(-\kappa\, b_{\theta,c}), \quad \kappa \geq 0, \tag{23}$$

so majority (larger $b_{\theta,c}$) have lower temperature, sharpening their competition and mitigating dominance. The loss becomes

$$\ell_{\text{sup}}^{\text{BiAL-Temp}} = -\log \frac{\exp(s_y(x)/\tau_y)}{\sum_c \exp(s_c(x)/\tau_c)}. \tag{24}$$

For instance discrimination (InfoNCE) you can analogously scale the positive/negative terms by $\tau$ chosen from the class of the anchor or by a mixture over candidate classes.

The SSL head (consistency/pseudo-labels) follows the same substitution: compute scores with $s^{\text{BiAL}}$ or $(s, \tau_c)$ and apply equation 19.

### A.4 Bias-aware FixMatch

Any pipeline that refines pseudo-labels (distribution alignment, confidence relabeling, FixMatch-style bias filtering) can be made bias-aware by using $E$ through the training process:

1. Propose labels from $p^E(a(u))$;

2. Apply your refinement rule (histogram alignment, confidence reweighting) on $p^E$;

3. Train the strong view on the refined label via $\ell_{\text{ssl}}(E(A(u)), \hat{q})$.

This uniformly reduces majority-preference in proposals and improves minority precision without adding extra heads.

## B Detailed Exposition of Theoretical Motivations

We formalize three guarantees for BiAL: (i) **Fisher consistency** for balanced error (BER) when decisions are made with debiased energy $E$; (ii) **dynamic regret** advantages under prior drift when the bias estimate tracks the current prior; and (iii) a **gradient analysis** showing how subtracting $b_\theta$ systematically suppresses majority bias and enlarges minority margins.

Throughout, let $K$ be the number of classes, $\eta_c(x) = \Pr(Y = c \mid X = x)$, and $\pi_c = \Pr(Y = c)$. For logits $z(x)$, we have defined the debiased energy in equation 3. With a centered bias vector $b_\theta \in \mathbb{R}^K$ (subtracting its mean to remove global shifts). Unless stated otherwise, we take $\beta = 1$ for decision rules; larger/smaller $\beta$ corresponds to cost re-scaling.

### B.1 Fisher consistency for BER

The equivalent of BER(equation 9) is:

$$\text{BER}(f) \;=\; \frac{1}{K} \sum_{c=1}^{K} \int \mathbf{1}\,(f(x) \neq c)\, p(x \mid Y = c)\, dx. \tag{25}$$

**Bayes rule for BER.** Using Bayes formula $p(x \mid Y = c) = \eta_c(x)\, p(x)/\pi_c$

$$\text{BER}(f) = \frac{1}{K} \int \left( \sum_c \tfrac{\eta_c(x)}{\pi_c} - \max_c \tfrac{\eta_c(x)}{\pi_c} \right) p(x)\, dx. \tag{26}$$

Thus the pointwise minimizer is equation 11, which we now connect to BiAL decisions.

**Assumption A (calibration).** The model is well-calibrated in the large-sample/realizable limit: $\text{softmax}(z(x)) \to \eta(x)$.

**Assumption B (bias consistency).** The bias estimate satisfies $b_\theta \to \log \pi + c\,\mathbf{1}$ (in probability), i.e., it recovers the log-prior up to a constant shift (irrelevant under softmax/argmax).

This holds, for example, if $b_\theta = \log \mathbb{E}_{I \sim \mathcal{I}}\, p_\theta(\cdot \mid I)$ with no-information inputs $\mathcal{I}$ that do not carry class cues and the learned classifier is calibrated on $\mathcal{D}_l \cup \mathcal{D}_u$; then $p_\theta(\cdot \mid I)$ converges to the model-induced prior which coincides with $\pi$ in the realizable limit.

**Decision rule of BiAL.** With $\beta = 1$,

$$f_{\mathrm{BiAL}}(x) = \arg \max_c E_c(x) = \arg \max_c \big(z_c(x) - b_{\theta,c}\big). \tag{27}$$

**Theorem 1 (Fisher consistency for BER).**

Under Assumptions AB.1 and BB.1, the BiAL decision rule $f_{\mathrm{BiAL}}$ is Fisher-consistent(Tasche, 2017) for BER, that is, $f_{\mathrm{BiAL}} \to f_{\mathrm{BER}}^\star$.

*Proof.* By Assumption A, $z_c(x) = \log \eta_c(x) + k(x)$ for some $k(x)$ independent of $c$. By Assumption B, $b_{\theta,c} = \log \pi_c + k'$ with $k'$ independent of $c$. Hence:

$$\arg \max_c \big(z_c(x) - b_{\theta,c}\big) = \arg \max_c \big(\log \eta_c(x) - \log \pi_c + k(x) - k'\big) = \arg \max_c \frac{\eta_c(x)}{\pi_c}, \tag{28}$$

which coincides with equation 11.

**Remark 1 (role of $\beta$).** If $\beta \neq 1$, the rule is Bayes-optimal for a cost-rescaled BER where the contribution of each class is weighted by $\pi_c^{-\beta}$. Thus, $\beta$ slightly ($\pm$) perturbs the effective operating point but keeps Fisher consistency w.r.t. the corresponding reweighted BER.

**Corollary 1 (approximate consistency).**

If $b_\theta = \log \pi + \delta$ with $\|\delta\|_\infty \leq \varepsilon$ and $z$ is calibrated, then the *excess* BER of $f_{\mathrm{BiAL}}$ over $f_{\mathrm{BER}}^\star$ is $O(\varepsilon)$ (proof via the dynamic regret bound below with $T = 1$).

### B.2 DYNAMIC REGRET UNDER PRIOR DRIFT

We consider training as a sequence of stages $t = 1, \ldots, T$ At stage $t$ the data-generating distribution has prior $\pi_t^\star$ and posterior $\eta^t(\cdot)$. Let $f_t$ be the classifier induced by BiAL using $b_t$ (the current bias estimate), and $f_t^\star$ be the stage-optimal BER Bayes rule that knows $\pi_t^\star$.

Define dynamic regret in BER:

$$\mathcal{R}_T = \sum_{t=1}^{T} \Big(\mathrm{BER}_t(f_t) - \mathrm{BER}_t(f_t^\star)\Big). \tag{29}$$

**Assumption C (bias tracking).** There exists $\varepsilon_t \geq 0$ such that

$$\| b_t - \log \pi_t^\star \|_\infty \leq \varepsilon_t. \tag{30}$$

**Lemma 1 (pointwise BER excess under prior error).**

Fix $t$ and $x$. Let $s_c(x) = \eta_c^t(x)/\pi_{t,c}^\star$, $\hat{s}_c(x) = \eta_c^t(x)/\exp\{b_{t,c}\}$.

Then

$$0 \leq \max_c s_c(x) - \max_c \hat{s}_c(x) \leq \big(e^{\varepsilon_t} - 1\big) \max_c s_c(x). \tag{31}$$

*Proof.* Since $|\log \pi_{t,c}^\star - b_{t,c}| \leq \varepsilon_t$, we have $e^{-\varepsilon_t} \leq \pi_{t,c}^\star/e^{b_{t,c}} \leq e^{\varepsilon_t}$. Thus for each $c$, $e^{-\varepsilon_t} s_c(x) \leq \hat{s}_c(x) \leq e^{\varepsilon_t} s_c(x)$.

Taking maxima gives $\max_c \hat{s}_c(x) \geq e^{-\varepsilon_t} \max_c s_c(x)$ and hence equation 31.

**Theorem 2 (dynamic regret bound).**

Under Assumptions AB.1 and CB.2,

$$\mathcal{R}_T \;\leq\; \frac{1}{K} \sum_{t=1}^{T} \left(e^{\varepsilon_t} - 1\right) \mathbb{E}_X \left[ \max_c \frac{\eta_c^t(X)}{\pi_{t,c}^\star} \right] \;\leq\; \frac{C}{K} \sum_{t=1}^{T} \varepsilon_t, \tag{32}$$

for a constant $C$ depending on $\pi_{\min} = \min_{t,c} \pi_{t,c}^\star$ (e.g., $C \leq e\,\pi_{\min}^{-1}$).

*Proof.* Using the BER decomposition and Lemma 1B.2,

$$\begin{aligned}
\mathrm{BER}_t(f_t) - \mathrm{BER}_t(f_t^\star) &= \frac{1}{K} \mathbb{E}_X \left[ \max_c s_c(X) - \max_c \hat{s}_c(X) \right] \\
&\leq \frac{1}{K} \left(e^{\varepsilon_t} - 1\right) \mathbb{E}_X \left[ \max_c s_c(X) \right].
\end{aligned} \tag{33}$$

Since $\max_c s_c(X) \leq \sum_c s_c(X) = \sum_c \eta_c^t(X)/\pi_{t,c}^\star \leq \pi_{\min}^{-1}$ , we obtain the second inequality and $\sum_t (e^{\varepsilon_t} - 1) \leq e \sum_t \varepsilon_t$ for $\varepsilon_t \in [0,1]$.

**Consequences.**

- If $b_t$ is an EMA/batch estimate whose error satisfies $\mathbb{E}\,\varepsilon_t = O(\sigma_t + \Delta_t)$ with sampling noise $\sigma_t$ and drift increment $\Delta_t = \| \log \pi_t^\star - \log \pi_{t-1}^\star \|_\infty$, then $\mathbb{E}\,\mathcal{R}_T = O(\sum_t (\sigma_t + \Delta_t))$.

- Any fixed prior $\bar{\pi}$ has $\varepsilon_t = \| \log \pi_t^\star - \log \bar{\pi} \|_\infty$, so $\sum_t \varepsilon_t$ scales at least linearly with total drift $\sum_t \Delta_t$ (triangle inequality). Hence tracking $b_t$ yields strictly smaller regret when drift is nontrivial.

**Corollary 2 (one-shot excess BER).**

In a single stage ($T = 1$), if $\|b - \log \pi^\star\|_\infty \leq \varepsilon$, then

$$\mathrm{BER}(f_{\mathrm{BiAL}}) - \mathrm{BER}(f_{\mathrm{BER}}^\star) \leq \frac{e}{K\,\pi_{\min}} \varepsilon. \tag{34}$$

### B.3 GRADIENT AND MARGIN EFFECTS OF SUBTRACTING $b_\theta$

We analyze how $E = z - \beta b_\theta$ redistributes probability mass and margins.

**Softmax sensitivity to $\beta$.**

Let $p^E = \mathrm{softmax}(E)$. Since $E = z - \beta b$ (we omit $\theta$ for brevity),

$$\frac{\partial p_i^E}{\partial \beta} = \sum_j \frac{\partial p_i^E}{\partial E_j} \frac{\partial E_j}{\partial \beta} = -\sum_j p_i^E (\delta_{ij} - p_j^E)\, b_j = p_i^E \big( \underbrace{\textstyle\sum_j p_j^E b_j}_{\mathbb{E}_{p^E}[b]} - b_i \big). \tag{35}$$

Thus, increasing $\beta$ decreases $p_i^E$ whenever $b_i > \mathbb{E}_{p^E}[b]$ (majority-biased classes) and increases $p_i^E$ when $b_i < \mathbb{E}_{p^E}[b]$ (minority-biased classes). This formally captures the "mass flows from high-bias to low-bias classes" effect.

**Gradients.** For supervised CE in a labeled pair $(x, y)$,

$$\nabla_z \ell_{\sup}^{\mathrm{BiAL}}(E, y) = p^E - e_y. \tag{36}$$

Compared to the baseline gradient $p - e_y$ (with $p = \mathrm{softmax}(z)$), equation 35 implies that as $\beta$ increases, the negative-class components $p_i^E$ for high-bias classes shrink, dampening their gradients; for the true class $y$, if $b_y$ is below the current average, $p_y^E$ increases, strengthening the positive

---

**Algorithm 2** Supervised BiAL (CE/LA/LDAM-compatible)

---

**Inputs:** labeled set $\mathcal{D}_l$; model $f_\theta$; epochs $T$; debias schedule $\{\beta_t\}_{t=1}^T$; (optional) margin mix weight $\{\lambda_t\}_{t=1}^T$.
**Output:** trained parameters $\theta$
  1: Initialize bias buffer $b \leftarrow \mathbf{0}$
  2: **for** $t = 1$ **to** $T$ **do**
  3:    **Estimate bias** $b_t$ *from a tiny set of no-information inputs*; apply centering and EMA to update $b$
  4:    **Form debiased energies** $E(x) \leftarrow z(x) - \beta_t b$ with $z(x) = f_\theta(x)$
  5:    **Compute supervised loss** on $E(x)$:
  6:       *CE/LA:* use standard cross-entropy on $\mathrm{softmax}(E)$ *(LA is a special case with fixed prior)*
  7:       *LDAM:* build bias-aware margins from $\pi^{(b)} = \mathrm{softmax}(b)$ and mix with count-based margins via $\lambda_t$; subtract on the true class
  8:    Update $\theta$ by minimizing the chosen loss on batches from $\mathcal{D}_l$
  9: **end for**
 10: **Return** $\theta$ *(test-time decision can use $E$ for BER-aligned prediction)*

---

gradient. The same form holds for the SSL cross-entropy on pseudo-labels (with $y = \hat{y}$) and for prototype/contrastive heads after replacing $z$ by the corresponding scores.

**Pairwise margins.** For any classes $a \neq c$,

$$\underbrace{E_a(x) - E_c(x)}_{\text{BiAL margin}} = \underbrace{z_a(x) - z_c(x)}_{\text{raw margin}} - \beta\,(b_a - b_c). \tag{37}$$

If $b_c > b_a$ (class $c$ more biased than $a$), then increasing $\beta$ enlarges the $a$–vs–$c$ margin by $\beta(b_c - b_a)$. In particular, for a minority class $a$ against a majority class $c$, equation 37 increases the minority's effective margins uniformly over inputs $x$.

**Theorem 3 (expected margin improvement for minorities).**

Let $y$ denote the ground-truth class and suppose $\mathbb{E}[b_y] \leq \mathbb{E}[b_c] - \Delta$ for all $c \neq y$ (minority gap $\Delta > 0$). Then for any $\beta > 0$,

$$\mathbb{E}\big[E_y(X) - \max_{c \neq y} E_c(X)\big] \;\geq\; \mathbb{E}\big[z_y(X) - \max_{c \neq y} z_c(X)\big] \;+\; \beta\,\Delta. \tag{38}$$

*Proof.* For each $x$, $E_y(x) - E_c(x) = z_y(x) - z_c(x) - \beta(b_y - b_c)$. Taking maximum over $c \neq y$ and expectations, and using $\max_c(u_c + v_c) \leq \max_c u_c + \max_c v_c$,

$$\mathbb{E}\big[\max_{c \neq y} E_c\big] \leq \mathbb{E}\big[\max_{c \neq y} z_c\big] - \beta\,\mathbb{E}\big[\min_{c \neq y}(b_y - b_c)\big]. \tag{39}$$

By the gap assumption, $\min_{c \neq y}(b_y - b_c) \leq -\Delta$ almost surely (or in expectation). Rearranging yields equation 38.

# C  METHODOLOGY IMPLEMENTATION DETAILS

## C.1  SUPERVISED IMPLEMENTATIONS

Supervised implementations follow the pseudocode by feeding $E$ into otherwise standard objectives. For CE/LA, this is simply CE on $E$ (equivalently, replacing the fixed $\tau \log \pi$ in LA with $\beta_t b_\theta$), with an optional logit debias $z \leftarrow z - \beta_t b_\theta$ during the ramp. For LDAM, we derive bias-aware, class-dependent margins from $\pi^{(b)} = \mathrm{softmax}(b_\theta)$ and mix them with the classic count-based margins using $\lambda_t$, then apply the usual true-class subtraction and scaling. Evaluation can be reported with the raw head or with the BiAL decision $E$, as noted in the main text.

### C.1.1   BIAL–LA (BIAS-AWARE LOGIT ADJUSTMENT)

To "bias-aware" the classic LA, simply replace $\tau \log \pi$ by $\beta b_\theta$ and (optionally) ramp $\beta$:

$$E(x) = z(x) - \beta_t\, b_\theta, \qquad \ell_{\mathrm{CE}}(\mathrm{softmax}(E), y). \tag{40}$$

This drop-in change can be used in place of CE anywhere CE/LA appears.

### C.1.2   BIAL–LDAM (BIAS-AWARE LDAM)

We combine optional logit debiasing with strength $\beta_t$, and bias-aware dynamic margins mixed with the standard LDAM margins via $\lambda_t$.

**(a) Logit debiasing.**

Before margin subtraction, replace logits by

$$z'(x) \;=\; z(x) - \beta_t\, b_\theta \quad \text{(stop-grad on } b_\theta). \tag{41}$$

This is the supervised analogue of using $E(x)$ mentioned earlier, with a ramped strength.

**(b) Bias-aware margins.**

Let $m_{\mathrm{std},c} \propto n_c^{-1/4}$ be standard LDAM margins from class counts. Define an effective prior $\pi^{\mathrm{eff}} = \mathrm{softmax}(b_\theta)$ and build bias-aware margins

$$m_{\mathrm{bias},c} \;=\; m_{\max} \cdot \frac{\left(\max_j \pi_j^{\mathrm{eff}}\right)^{\beta_{\mathrm{BA}}}}{(\pi_c^{\mathrm{eff}})^{\beta\_\mathrm{BA}}} \;\;\propto\;\; (\pi_c^{\mathrm{eff}})^{-\beta_{\mathrm{BA}}} \quad \text{(clamped to } [m_{\min}, m_{\max}]). \tag{42}$$

We mix them as

$$m_c(t) \;=\; (1 - \lambda_t)\, m_{\mathrm{std},c} \;+\; \lambda_t\, m_{\mathrm{bias},c}, \tag{43}$$

and subtract $m_c(t)$ from the true-class logit only, then apply the usual LDAM scaling $s$. This realizes equation 20 and equation 21 with a smooth transition from the vanilla LDAM to its bias-aware form.

## C.2   SEMI-SUPERVISED IMPLEMENTATIONS

We adopt a FixMatch-style structure (weak/strong views, confidence threshold), but generate and train on pseudo-labels from debiased energy $E$, not raw logits $z$. The same principle applies to CCL heads and other pipelines.

### C.2.1   BIAL–FIXMATCH (CE BRANCH)

For each unlabeled $u$, compute

$$E(a(u)) = z(a(u)) - \beta_t b_\theta, \qquad p^E(a(u)) = \mathrm{softmax}(E(a(u))). \tag{44}$$

If $\max_c p_c^E(a(u)) \geq \tau_{\mathrm{pl}}$, set $\hat{y} = \arg\max_c p_c^E(a(u))$ and train the strong view with

$$\ell_{\mathrm{ssl}} = \mathrm{CE}\big(\mathrm{softmax}(E(A(u))),\ \mathrm{one-hot}(\hat{y})\big). \tag{45}$$

This aligns both proposal and learning with the BiAL correction. Confidence thresholding becomes substantially more robust for tail classes once majorities are down-weighted by $b_\theta$.

### C.2.2 BIAL–CCL (CONTRASTIVE HEAD)

Two equivalent plug-ins:

- **Debiased class scores.** For prototype-based logits $s_c(x)$, use $cs_c^{\text{BiAL}}(x) = s_c(x) - \beta_t b_{\theta,c}$ in both supervised and SSL heads (InfoNCE softmax is unchanged).
- **Class-wise temperatures.** Use $\tau_c = \tau_0 \exp(-\kappa b_{\theta,c})$ and feed $s_c(x)/\tau_c$ to the softmax. Majority classes (large $b_{\theta,c}$) get smaller temperatures, effectively sharpening competition and mitigating over-dominance; minorities get larger temperatures, easing positive alignment.

### C.3 OPTIONAL REGULARIZERS AND SCHEDULES

- **Bias smoothing/variance control:** $\Omega(b_\theta) = \|b_\theta\|_2^2$ or $\mathrm{Var}(b_\theta)$ to avoid extreme corrections early; EMA already serves as implicit regularization.
- **Strength scheduling:** $\beta_t \uparrow$ from small to moderate (warm-up), or adapt $\beta_t$ to the entropy/variance of $b_\theta$.

## D EXPERIMENTAL SETUP

### D.1 TRAINING DATASETS

We use a variety of commonly adopted SSL datasets to conduct our experimental analysis, including CIFAR-10-LT, CIFAR-100-LT, STL10-LT and ImageNet-127 in different ratios $\gamma$ . To create imbalanced versions of the datasets, the labeled set $\mathcal{D}_l$ is made long-tailed by class-wise exponential decay $n_c^L = n_{\max} \cdot \gamma^{-\frac{c-1}{K-1}}$ ,with classes ordered by frequency. For CIFAR10/100-LT, We carry out experiments under three regimes like recent LTSSL works: Consistent, Uniform, Reverse. Experiments are conduted with all comparison methods in settings where $N1 = 500, M1 = 4000$, and $N1 = 1500, M1 = 3000$. We adopt imbalance ratios of $\gamma_l = \gamma_u = 100$ and $\gamma_l = \gamma_u = 150$ for consistent settings, while for uniform and reversed settings, we adopt $\gamma_l = 100, \gamma_u = 1$ and $\gamma_l = 100, \gamma_u = 1/100$, respectively. Given the absence of ground-truth labels for the unlabeled data of the STL10-LT dataset, we manage the experiments by adjusting the imbalance ratio of the labeled data, where we set the labeled imbalance ratio of $\gamma_l = 10$ or $\gamma_l = 20$. And for ImageNet-127, created by ImageNet(Russakovsky et al., 2015), we experiment under consistent settings on images down-sampled to $32 \times 32$ and $64 \times 64$.

### D.2 IMPLEMENTATION DETAILS

Our experimental configuration largely aligns with Fixmatch and CCL. Specifically, we apply the WideResNet-28-2(Zagoruyko & Komodakis, 2016) architecture to implement our method on the CIFAR10-LT, CIFAR100-LT and STL10-LT datasets; and ResNet-50(He et al., 2016) on ImageNet-127. The performance evaluation of these methods is based on the top-1 accuracy metric on the test set. We present the mean and standard deviation of the results from three independent runs for each method.

For FixMatch based methods, our BiAL-FixMatch uses the Adam optimizer(Kingma, 2014). We used the EMA of the network parameters for each iteration to evaluate the classification performance. We used random cropping and horizontal flipping for weak data augmentation and Cutout(DeVries & Taylor, 2017) and RandomAugment(Cubuk et al., 2020) for strong data augmentation. We set the mini batch size to 32, relative size of the unlabeled to labeled mini-batches $\mu$ to 2, and learning rate of the optimizer to $1.5 \times 10^{-3}$. We trained BiAL-FixMatch for 500 epochs, where 1 epoch = 500 iterations. For the experiments on CIFAR 100, we set the weight decay parameter of L2 regularization (for EMA parameters) to 0.08. For the experiments on CIFAR-10, STL-10, and ImageNet-127, we set the weight decay parameter of L2 regularization to 0.04.

For CCL based methods, our BiAL-CCL keep the settings same as original model. We adopt the common training paradigm that the network is trained with standard SGD(momentum 0.9, weight decay $5 \times 10^{-4}$)(Polyak, 1964; Sutskever et al., 2013) for 500 epochs, where each epoch consists of 500 mini-batches, and a batch size of 64 for both labeled and unlabeled data. We use a cosine

learning rate decay(Loshchilov & Hutter, 2016) where the initial rate is 0.03, we set $\tau = 2.0$ for logit adjustment on all datasets, except for ImageNet-127, where $\tau = 0.1$. We set the temperature $T = 1$ and the threshold $\zeta = -8.75$ for the energy score following(Yu et al., 2023), and we set $\lambda_1 = 0.7$, $\lambda_2 = 1.0$ on CIFAR10/100-LT and $\lambda_1 = 0.7$, $\lambda_2 = 1.5$ on STL10-LT and ImageNet-127 datasets for the final loss.

For the parameters in our method, we mainly make the following settings: For BiAL-FixMatch, we set default $E_{\text{warm}} = 50$ epochs and $E_{\text{ramp}} = 20$, with $\beta = 1.0$ on CIFAR10/100-LT and STL-10, while $\beta = 0.1$ on ImageNet-127. For BiAL-CCL, we set default $E_{\text{warm}} = 50$ epochs and $E_{\text{ramp}} = 20$, with $\beta = 0.5$ on CIFAR10/100-LT and STL-10, while $\beta = 0.03$ on ImageNet-127.

In addition, our method is implemented using the PyTorch library and experimented on NVIDIA RTX 3090s.

# E  FURTHER ANALYSIS

## E.1  SEMI-SUPERVISED LEARNING

Figure 5 presents the confusion matrix comparing CCL and BiAL-CCL in three different experimental settings. We test both models on CIFAR10-LT dataset under consistent($\gamma_l = \gamma_u = 150$), uniform($\gamma_l = 100, \gamma_u = 1$) and reverse($\gamma_l = 100, \gamma_u = 1/100$) settings. As we can see in the figure, BiAL can help CCL care more about tail classes without sacrificing the performance of head classes. Especially for tail categories, such as category 9, BiAL-CCL shows a significant improvement compared to CCL and also achieves higher overall accuracy.

Furthermore, we employ the t-distributed stochastic neighbor embedding (t-SNE)(Maaten & Hinton, 2008) to visualize the representations learned by both methods. As mentioned above, the comparative results on are depicted in Figure 6 under consistent, uniform, and reverse settings. The figure demonstrates that BiAL enables CCL provide more distinct classification boundaries. Especially for the setting of uniform and reverse, when the distribution of unlabeled data is inconsistent with that of labeled data, BiAL can more accurately grasp the deviation of the model and achieve better performance.

To further examine whether BiAL is compatible with other dual branch methods, we also integrate it into ACR and report the results in Table6. In this experiment, we simply replace the logits used in ACR correction with our debiased energies $E(x) = z(x) - \beta_t b_\theta$, while keeping all other training details and hyperparameters unchanged. We evaluate under the consistent setting on CIFAR10-LT and CIFAR100-LT , following the same protocols as in the main experiments. Across all configurations, BiAL-ACR consistently improves over FixMatch+ACR, and it also outperforms FixMatch combined with CDMAD, LCGC, or CPE. The improvements are still clear in the more challenging CIFAR100-LT scenarios, where the pseudo-labeling induces strong drift over training. These results indicate that BiAL provides a complementary, rather than redundant, bias correction to ACR: while ACR leverages fixed priors within an energy-based framework, replacing these priors with the model-induced bias allows the correction to better track the evolving effective prior, yielding more robust performance under long-tailed semi-supervised learning.

Table 6: Test accuracy in consistent setting on CIFAR10-LT and CIFAR100-LT datasets for BiAL-ACR. The best results are in **bold**.

| | CIFAR10-LT | | | | CIFAR100-LT | | | |
|---|---|---|---|---|---|---|---|---|
| | $\gamma_l = \gamma_u = 100$ | | $\gamma_l = \gamma_u = 150$ | | $\gamma_l = \gamma_u = 10$ | | $\gamma_l = \gamma_u = 20$ | |
| Algorithm | $N_1 = 500$ $M_1 = 4000$ | $N_1 = 1500$ $M_1 = 3000$ | $N_1 = 500$ $M_1 = 4000$ | $N_1 = 1500$ $M_1 = 3000$ | $N_1 = 50$ $M_1 = 400$ | $N_1 = 150$ $M_1 = 300$ | $N_1 = 50$ $M_1 = 400$ | $N_1 = 150$ $M_1 = 300$ |
| FixMatch + CDMAD(Lee & Kim, 2024) | 80.3±0.21 | 83.6±0.46 | 73.3±0.63 | 80.5±0.76 | 50.6±0.44 | 60.3±0.32 | 44.7±0.14 | 54.3±0.44 |
| FixMatch + LCGC(Xing et al., 2025) | 81.2±0.73 | 83.9±0.36 | 74.3±1.92 | 80.8±0.32 | 50.9±0.45 | 60.2±0.57 | 44.6±0.81 | 55.3±0.48 |
| FixMatch + ACR(Wei & Gan, 2023) | 81.6±0.19 | 84.1±0.39 | 77.0±1.19 | 80.9±0.22 | 51.1±0.32 | 61.0±0.41 | 44.3±0.21 | 55.2±0.28 |
| FixMatch + CPE(Ma et al., 2024) | 80.7±0.96 | 84.4±0.29 | 76.8±0.53 | 82.3±0.34 | 50.3±0.34 | 59.8±0.16 | 43.8±0.28 | 55.6±0.15 |
| w/ BiAL-ACR | **82.1±0.54** | **85.6±0.48** | **78.4±0.44** | **82.3±0.31** | **52.1±0.67** | **62.4±0.53** | **45.1±0.42** | **56.5±0.31** |

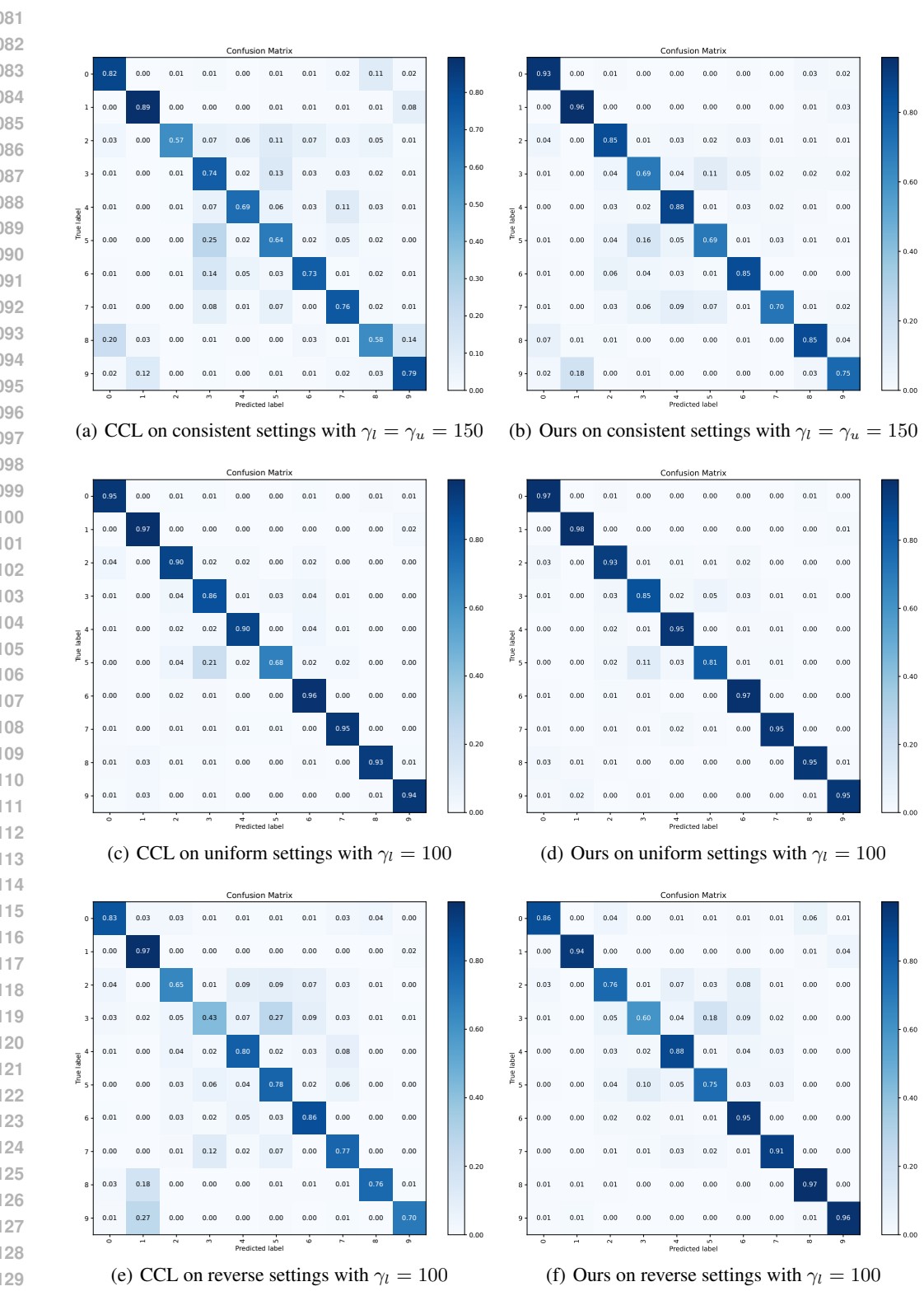

(a) CCL on consistent settings with $\gamma_l = \gamma_u = 150$

(b) Ours on consistent settings with $\gamma_l = \gamma_u = 150$

(c) CCL on uniform settings with $\gamma_l = 100$

(d) Ours on uniform settings with $\gamma_l = 100$

(e) CCL on reverse settings with $\gamma_l = 100$

(f) Ours on reverse settings with $\gamma_l = 100$

Figure 5: The confusion matrices of the predictions on the test set of CIFAR-10-LT dataset in three different settings for CCL and BiAL-CCL.

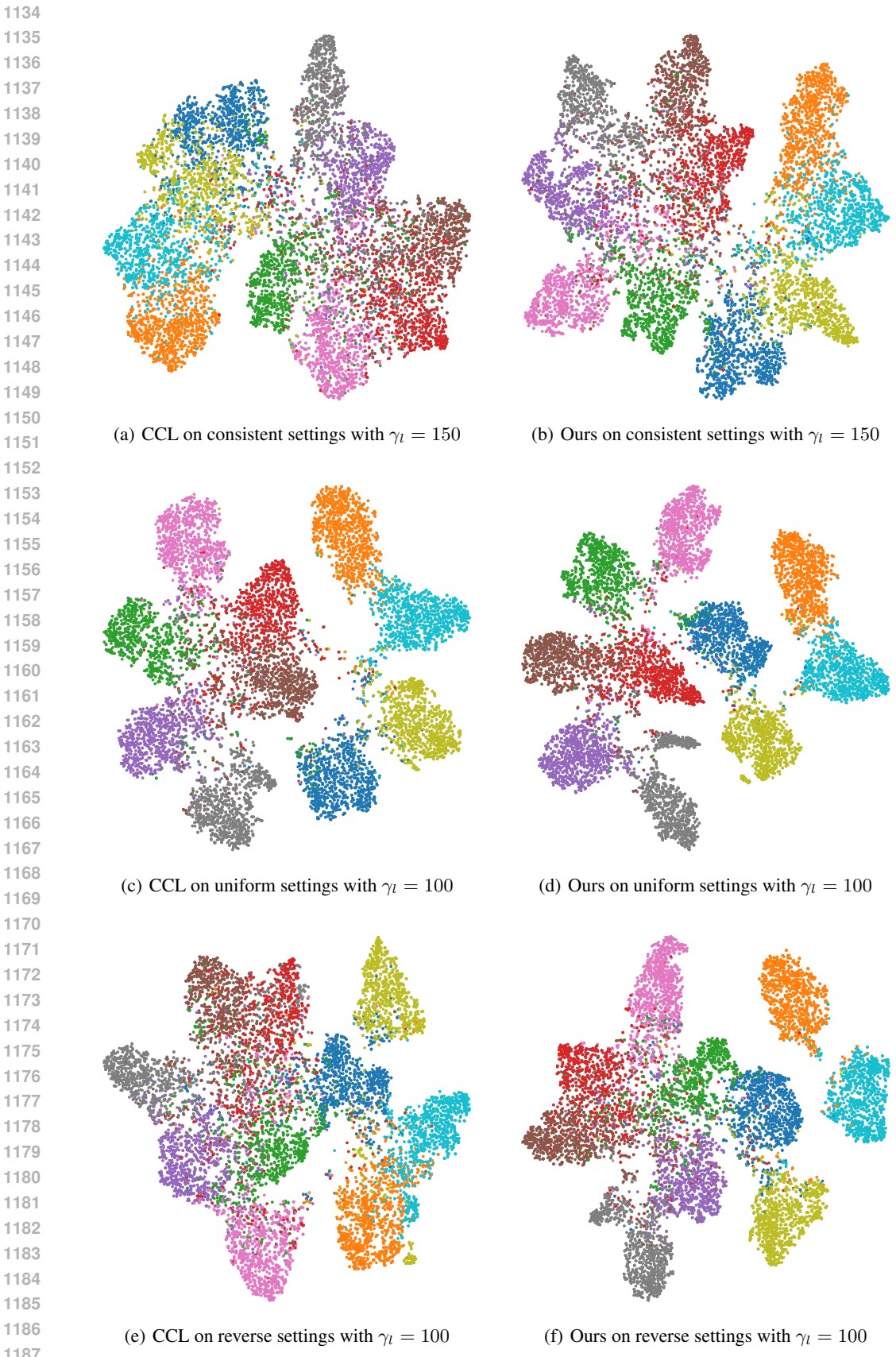

(a) CCL on consistent settings with $\gamma_l = 150$     (b) Ours on consistent settings with $\gamma_l = 150$

(c) CCL on uniform settings with $\gamma_l = 100$     (d) Ours on uniform settings with $\gamma_l = 100$

(e) CCL on reverse settings with $\gamma_l = 100$     (f) Ours on reverse settings with $\gamma_l = 100$

Figure 6: The t-SNE visualization of the test set for CCL and BiAL-CCL on CIFAR-10-LT dataset in three different settings.

### E.2 Supervised Learning

After introducing our method into supervised learning, we compared it with the baseline through experiments and verified the effectiveness of the method.

For supervised learning, we instantiate BiAL by replacing logits $z$ with debiased energies $E(x) = z(x) - \beta_t b_\theta$ inside otherwise standard objectives. For CE/LA, the fixed prior term $\tau \log \pi$ is substituted by $\beta_t b_\theta$. For LDAM, we derive bias-aware class margins from the effective prior $\pi_{\text{eff}} = \text{softmax}(b_\theta)$ and mix them with standard count-based margins via a weight $\lambda_t$; the mixed margin is subtracted on the true class followed by the usual LDAM scaling, yielding a smooth transition from vanilla to bias-aware LDAM.

We evaluate BiAL in the fully supervised regime on CIFAR10-LT and CIFAR100-LT under the consistent long-tailed setting and report results in Table 7. All methods use the same backbone and ResNet50 on CIFAR datasets.

Table 7: Test accuracy in consistent setting on CIFAR10-LT and CIFAR100-LT datasets. The best results are in **bold**.

| | CIFAR10-LT | | CIFAR100-LT | |
|---|---|---|---|---|
| Algorithm | $\gamma_l = \gamma_u = 100$ | $\gamma_l = \gamma_u = 150$ | $\gamma_l = \gamma_u = 10$ | $\gamma_l = \gamma_u = 20$ |
| Supervised | 47.3±0.95 | 44.2±0.33 | 29.6±0.57 | 25.1±1.14 |
| w/ LA | 53.3±0.44 | 49.5±0.40 | 30.2±0.44 | 26.5±1.31 |
| w/ BiAL-LA | 55.6±0.71 | 52.1±0.62 | 31.7±0.83 | 28.2±0.98 |
| w/ LDAM | 74.4±0.78 | 70.7±0.63 | 55.7±0.20 | 51.4±0.44 |
| w/ LDAM-DRW | 77.5±0.60 | 73.9±0.50 | 57.3±0.53 | 53.6±0.34 |
| w/ BiAL-LDAM | 76.0±0.73 | 72.3±0.43 | 56.8±1.14 | 52.7±0.83 |
| w/ BiAL-LDAM-DRW | **79.2±0.81** | **76.0±0.64** | **59.1±0.42** | **54.3±0.91** |

### E.3 Sensitive analysis of hyperparameters

As described in Table 8, BiAL is relatively robust to the fluctuation of $\beta$ from 0.5 to 1.0 for FixMatch and from 0.1 to 0.5 for CCL. However, when $\beta$ is set to 1.5 or even larger, it amplifies debiasing, resulting in a performance decrease. When $\beta$ is set to be smaller than 0.1, it mitigates overcorrection.

Table 8: Sensitive analysis of $\beta$ under consistent setting of CIFAR10/100-LT

| | BiAL-FixMatch | | BiAL-CCL | |
|---|---|---|---|---|
| $\beta$ | CIFAR-10 | CIFAR-100 | CIFAR-10 | CIFAR-100 |
| 0.1 | 83.1 | 54.6 | 86.0 | 57.7 |
| 0.5 | 83.2 | 55.0 | **86.1** | **57.8** |
| 1.0 | **83.9** | **55.2** | 85.9 | 57.8 |
| 1.5 | 81.7 | 54.1 | 84.8 | 55.7 |
| 2.0 | 80.3 | 53.5 | 85.1 | 55.6 |

Table 9 investigates the effect of the warm-up length $E_{\text{warm}}$ and ramp length $E_{\text{ramp}}$ in BiAL. Overall, the results show that BiAL is fairly insensitive to these hyperparameters: changing $E_{\text{warm}}$ and $E_{\text{ramp}}$ within a reasonable range only leads to minor fluctuations, and all configurations still outperform the corresponding base SSL framework. For warm-up stage $E_{\text{warm}}$, we follow a standard design principle which aims to avoid introducing debiasing too early, when the backbone has not yet learned basic discriminative structure and the estimated bias is dominated by noise. In practice, $E_{\text{warm}}$ does not need to be carefully tuned: as long as debiasing is activated after the model has acquired a reasonable classification ability and there is sufficient time before the end of training, the exact value has little impact. The ramp stage $E_{\text{ramp}}$ mainly serves as a smooth transition from no debiasing to the full bias-aware regime, preventing sudden shifts in the effective decision boundary that could cause optimization oscillations. These observations justify our simple piecewise-linear schedule and indicate that BiAL does not rely on delicate tuning of $E_{\text{warm}}$ and $E_{\text{ramp}}$.

Table 9: Sensitive analysis of $E_{\text{warm}}$ and $E_{\text{ramp}}$ under consistent setting of CIFAR10/100-LT

| $E_{\text{warm}}$ | CIFAR-10 | CIFAR-100 | $E_{\text{ramp}}$ | CIFAR-10 | CIFAR-100 |
|---|---|---|---|---|---|
| 30 | 85.9 | 63.5 | 10 | 86.0 | 63.7 |
| 50 | **86.4** | **63.9** | 20 | **86.4** | **63.9** |
| 80 | 86.3 | 64.9 | 30 | 86.3 | 63.8 |
| 100 | 86.4 | 63.7 | 40 | 86.3 | 63.7 |
| 150 | 86.2 | 63.7 | 50 | 86.2 | 63.8 |

### E.4 TIME COMPLEXITY ANALYSIS

We measure complexity relative to the forward/backward passes of the backbone, which are the dominant cost. Let $B$ be the batch size, $K$ the number of classes, $D$ the feature dimension, $|\mathcal{B}_I|$ the mini-batch size of no-information inputs used to probe model bias, and $E_{\text{est}}$ the refresh cadence.

BiAL adds only two lightweight operations on top of any baseline: (i) a class-length subtraction per sample to form debiased energies $E = z - \beta b_\theta$, which costs $\mathcal{O}(BK)$ per step, and (ii) a low-frequency, amortized bias refresh that forwards $|\mathcal{B}_I|$ no-information inputs every $E_{\text{est}}$ steps and aggregates them, contributing $|\mathcal{B}_I|/E_{\text{est}}$ single-sample forwards plus $\mathcal{O}(|\mathcal{B}_I|/E_{\text{est}} \cdot K)$ bookkeeping per step. Both do not strictly alter the asymptotic order of the baseline.

Concretely, BiAL-FixMatch preserves the baseline loss-side complexity $\mathcal{O}(BD + BK)$ (or $\mathcal{O}(BK)$ when no projector is used); the added $\mathcal{O}(BK)$ and the tiny amortized refresh do not change the order. For BiAL-CCL, the dominant terms remain those of CCL, which yields $\mathcal{O}(B^2D + B^2K + B^3)$. Thus, BiAL is plug-and-play and does not change the asymptotic training complexity of FixMatch or CCL; it only adds a negligible linear-time adjustment and a low-frequency amortized probe.

Table 10: Average batch time of each algorithm.

| Algorithm | CIFAR-10 | CIFAR-100 | STL-10 |
|---|---|---|---|
| CCL | 0.173 sec/iter | 0.175 sec/iter | 0.231 sec/iter |
| BiAL-CCL | 0.175 sec/iter | 0.176 sec/iter | 0.243 sec/iter |

### E.5 EFFECT OF DIFFERENT NO-INFORMATION BASELINES

In our main experiments, the bias vector $b_t$ in BiAL is estimated using a batch of "no-information" inputs instantiated as all-black images. To assess the robustness of the proposed bias estimation scheme and test whether the performance of BiAL is sensitive to this particular choice of baseline input, we conduct an ablation study on CIFAR10-LT with BiAL-FixMatch and BiAL-CCL under the same configuration as in Tab. 1, varying only the input of the baseline image used to compute $b_t$.

Specifically, we consider nine types of no-information inputs, as shown in Tab. 11. For each baseline type, all other training hyperparameters and random seeds are kept fixed. Across all nine probe types, the test performance of BiAL-FixMatch on CIFAR10-LT remains very similar, with only minor fluctuations in accuracy. This observation indicates that BiAL does not rely on a specific color or pattern for the no-information inputs; instead, it is robust to the exact instantiation of different baselines used for bias estimation.

Overall, the ablation in Tab. 11 shows that BiAL is largely insensitive to the exact choice of no-information baseline, as long as the probe does not introduce strong high-frequency noise. For all constant-color images (black, gray, red, green, blue, white), the test accuracy of BiAL-FixMatch and BiAL-CCL stays within a narrow band, and the black baseline is only marginally better than other solid colors. This is consistent with our bias-estimation design: when the input is spatially constant, early convolution and normalization layers remove most absolute-intensity differences and the classifier is effectively probed by a "featureless" input, so the estimated bias $b_t$ mainly reflects the model's intrinsic class preference rather than the specific RGB value. In contrast, high-variance Gaussian noise yields a clearly inferior probe, as the random high-frequency patterns excite filters in an unstable and input-dependent way, increasing the variance of the bias estimate and thus the

Table 11: Effect of different no-information baseline images on CIFAR10-LT with BiAL-FixMatch and BiAL-CCL.

| Probe type | RGB value / description | BiAL-FixMatch | BiAL-CCL |
|---|---|---|---|
| Black | $(0, 0, 0)$ | 83.9 | 86.5 |
| Gray | $(128, 128, 128)$ | 83.5 | 86.3 |
| Red | $(255, 0, 0)$ | 82.9 | 86.1 |
| Green | $(0, 255, 0)$ | 83.7 | 86.2 |
| Blue | $(0, 0, 255)$ | 83.9 | 86.4 |
| White | $(255, 255, 255)$ | 83.2 | 86.3 |
| Gaussian noise | low-variance noise, clipped to valid range | 74.5 | 82.1 |
| Gaussian-filtered | Gaussian-blurred random patterns | 80.1 | 84.7 |
| Non-image | $(511, 511, 511)$ (out-of-range constant) | 83.0 | 86.2 |

mismatch between $b_t$ and the effective pseudo-label prior $\tilde{\pi}_t^{\mathrm{PL}}$, which our theory links to higher balanced error and regret. Gaussian-filtered noise, which suppresses these high-frequency fluctuations, sits between the two extremes, while the out-of-range "non-image" constant behaves similarly to other constant baselines, further indicating that BiAL only requires structurally uninformative probes rather than a particular color. Taken together, these results empirically confirm that BiAL does not rely on a specific baseline color and that potential correlations between certain colors and object classes (such as "blue = sky") do not materially affect its debiasing behavior.

# F COMPARISON WITH OTHER METHODS

## F.1 COMPARISON WITH CDMAD

CDMAD, a post-hoc score correction that leaves baseline losses optimizing raw logits, subtracts a bias vector estimated from non-informative inputs mainly for pseudo-labeling and test inference. And BiAL internalizes debiasing as a unified training objective: it systematically replaces logits $z$ with bias-aware energies $E_t = z - \beta_t b_t$ in all modules, including supervised CE/LDAM, unlabeled consistency and pseudo-labels, contrastive/prototype heads, and test-time prediction, thereby enforcing a single, consistent decision rule throughout training and inference. Methodologically, BiAL tracks the epoch-varying effective prior induced by SSL via a centered log-mean-exp bias estimator updated with EMA and governed by a warm-up/ramp on $\beta_t$, providing stability and controllability absent in CDMAD's one-shot subtraction. This objective-level integration also makes BiAL loss-family compatible and theoretically cleaner by reducing prior-mismatch regret as the label distribution drifts, while retaining strict generality. With appropriate choices of the bias estimate $b_t$, strength $\beta_t$, and where the correction is applied, BiAL can reproduce the decision behavior of CDMAD and LA, while extending beyond them when the same bias-aware energy is used uniformly across training and inference. In practice, BiAL achieves these gains with negligible overhead, delivering stronger end-to-end consistency, better stability under drift, and broader extensibility than CDMAD.

## F.2 COMPARISON WITH DEBIASPL

Both methods correct class-prior deflection via a class-wise additive term before softmax, but they differ in what is estimated, where it is applied, and how broadly it shapes training. DebiasPL builds a data-driven dynamic prior from unlabeled predictions, using EMA of the model's marginal over unlabeled data $\log \hat{p}$, and subtracts it mainly for pseudo-labeling, often paired with a $\hat{p}$-aware margin; supervised CE typically remains on raw logits. BiAL instead measures a model-intrinsic bias via non-informative probes $b_t$ and internalizes debiasing at the objective level, replacing logits by bias-aware energy $E_t = z - \beta_t b_t$ everywhere, yielding a single, train–test-consistent decision rule. Practically, BiAL offers finer stability controls and broader compatibility/extensibility, and it separates inherent model bias from data-distribution effects, which helps reduce prior-mismatch regret under drifting effective priors in SSL. Notably, BiAL can reproduce DebiasPL-style behavior by setting $b_t \leftarrow \log \hat{p}$ and applying $E_t$ only to pseudo-labels/test, but it extends beyond DebiasPL by

turning debiasing from a localized correction into a unified, stable, and end-to-end training objective with negligible extra compute.

## G    THE USE OF LARGE LANGUAGE MODELS

In this paper, the use of LLM mainly exists in the polishing of the article and the adjustment of some table formats.

