# OpenReview forum: "How much correction is adequate? A Unified Bias-Aware Loss for Long-Tailed Semi-Supervised Learning"
_ICLR.cc/2026/Conference — Submitted to ICLR 2026_

### Official Review · Reviewer_waeb · 2025-10-29

**Soundness:** 2
**Presentation:** 2
**Contribution:** 2
**Rating:** 2
**Confidence:** 4

**Summary:**

This paper attempts to address the issue of dynamic bias in long-tailed semi-supervised learning (LTSSL). The authors propose BiAL, which replaces the static class prior with an online estimate of the model's own bias, measured from its output on no-information inputs. The authors subtract this estimated bias from the model's logits to form debiased energy, and uniformly applying this energy across all loss functions provides a more adaptive and effective correction mechanism. The experiments reportedly achieve state-of-the-art performance on multiple datasets.

**Strengths:**

1. The idea of probing model bias with no-information inputs seem to be simple and effective. And the experiments verified that the method achieve sota performance.
2. The paper writing is clear and easy to understand.

**Weaknesses:**

1. The core idea of this paper that using the model's response to no-information inputs to estimate and correct for its bias has been proposed in CDMAD [1]. The use of bias just dynamic variant of the classic Logit Adjustment (LA). [2]
2. Lack theoretical foundation for why $b_{\theta}$ is a good estimator for the "effective training prior. The paper provides no rigorous proof or analysis whatsoever.
3. The paper is replete with grandiose terms like unified, universal, and "fundamental," which do not align with the actual substance of the contribution. The so-called "unification" is just applying a simple logit subtraction to different components.
[1] Cdmad: class-distribution-mismatch-aware debiasing for classimbalanced semi-supervised learning.
[2] Long-tail learning via logit adjustment.

**Questions:**

See in Weaknesses.

---

> ### Author Response · Authors · 2025-11-21
>
> We sincerely thank you for carefully reading our paper and for the detailed, thought-provoking comments. And below we address the three main concerns.
>
> Q1. The core idea has been proposed in CDMAD and just dynamic variant of LA
>
> We agree that BiAL is conceptually related to CDMAD[1] and LA[2] in that all of them exploit model bias to correct imbalanced predictions. However, our goal is not merely to subtract yet another vector from logits, but to (i) define a single debiased energy that is used consistently in all components of long-tailed SSL pipelines, and (ii) analyze this design under a dynamic prior–drift perspective, which is different in both mechanism and scope from stage-wise post-hoc corrections.
>
> In CDMAD and standard LA, bias correction is applied only to specific modules. LA uses a static label prior and adjusts logits in the supervised head, while pseudo-labeling and contrastive objectives still operate on the original logits. CDMAD estimate a bias from “no-information” or low-content inputs and primarily use it to correct pseudo-labels and test scores, again leaving the supervised and representation losses unchanged. By contrast, BiAL defines a debiased energy $E_t(x) = z_t(x) - \beta_t \ b_t,$ where $b_t$ is a model-internal bias estimated from no-information probes, and uses this same $E_t$ for the labeled loss (CE/LA, with or without LA), the unlabeled consistency and pseudo-label losses, the contrastive/prototype heads, and test prediction. All decision rules during training and inference are thus expressed in a shared bias-aware energy space, rather than mixing original logits and corrected logits at different stages. We have clarified this point in Sec. 3 and in App. F.1, where we now explicitly contrast BiAL with CDMAD and LA.
>
> The different behavior also appears clearly in experiments. We further examined what happens if CDMAD is plugged into strong multi-head LA pipelines such as CCL[3] and ACR[4]. If BiAL were essentially “CDMAD applied everywhere”, then CCL+CDMAD should behave similarly to BiAL-CCL. Instead, in the additional experiments, we observe that ACR+CDMAD and CCL+CDMAD obviously degrades performance (for example, on CIFAR-10-LT the top-1 accuracy drops from 84.5\% for CCL to 78.7\% for CCL+CDMAD, and a similar drop occurs on CIFAR-100-LT), whereas BiAL-CCL consistently matches or improves CCL under the same settings. Moreover, BiAL-ACR yields 1–2\% overall gains and larger improvements on tail classes without changing the ACR architecture (App. E.1). These results suggest that simply stacking a post-hoc debiaser like CDMAD on top of an already LA-heavy multi-head pipeline leads to double debiasing and breaks the original probability structure, while replacing logits with debiased energies inside the losses, as done by BiAL, keeps the whole pipeline coherent and stable.
>
> On the theoretical side, Sec. 3.5 and App. B analyze BiAL in a Bayes error and dynamic regret framework. We formalize the effective prior $\tilde{\pi}_t^{PL}$ induced by accepted pseudo-labels, show that a fixed label prior can incur excess Bayes error and cumulative dynamic regret when $\tilde{\pi}_t^{PL}$ drifts under long-tailed SSL, and derive bounds indicating that the bias-aware energy $E_t$ reduces dynamic regret compared to static-prior LA when $b_t$ approximates $\log \tilde{\pi}_t^{PL}$. To the best of our knowledge, such a dynamic prior–drift analysis, explicitly contrasting static priors with bias-aware energies, is not present in CDMAD or standard LA.
>
> We fully acknowledge that BiAL is related in spirit to LA- and CDMAD-style corrections, and that these methods can be recovered as special cases in our energy formulation (for example, a static $b_t$ with $\beta_t=1$ reduces to LA in the supervised head, and a one-shot correction at test time resembles CDMAD). Our intention is precisely to provide a more complete and systematic framework that unifies such corrections at the loss level across all heads, remains compatible with strong multi-head LA pipelines such as FixMatch, CCL, and ACR, and is supported by a dynamic prior–drift analysis. We have clarified this positioning in Sec. 2 and App. F.1.
>
> [1] Class-Distribution-Mismatch-Aware Debiasing for Class-Imbalanced Semi-Supervised Learning. CVPR 2023
>
> [2] Long-tail learning via logit adjustment.
>
> [3] Continuous Contrastive Learning for Long-Tailed Semi-Supervised Recognition. Neurips 2024
>
> [4] Towards Realistic Long-Tailed Semi-Supervised Learning: Consistency Is All You Need. CVPR 2023

---

> ### Author Response · Authors · 2025-11-21
>
> Q2. Lack theoretical foundation
>
> We are grateful for this comment and apologize if our presentation made the theoretical aspects insufficiently visible. The current version already contains a theoretical analysis in Sec. 3.5 and App. B. There we connect BiAL to the Bayes-optimal decision rule under a time-varying effective prior $\tilde{\pi}_t^{PL}$, show that using a static label prior leads to excess Bayes error and cumulative dynamic regret when $\tilde{\pi}_t^{PL}$ drifts, and prove that, under mild assumptions, a classifier using the debiased energy $E_t = z_t - \beta_t b_t$ is Fisher-consistent and converges to the Bayes rule when $b_t$ approximates $\log \tilde{\pi}_t^{PL}$.
>
> Regarding the reasonableness of $b_t$ as a bias estimator, our design and experiments provide supporting evidence. By construction, $b_t$ is obtained from no-information probes via log-mean-exp and an EMA update (Sec. 3.1 and Sec. 4.1), so it captures a model-internal bias that is independent of specific image content and reflects the model’s current preference over classes, which is precisely the type of quantity we need as an effective prior. New ablations in App. E.5 show that BiAL is robust to the choice of low-information probe: replacing black images with gray, white, colored, constant non-image inputs, or blurred patterns yields almost identical performance, whereas high-frequency Gaussian noise (which violates the “low-information” assumption) significantly degrades performance. This suggests that $b_t$ is not tied to a particular color or pattern, but indeed reflects a stable model-induced bias.
>
> Q3. Grandiose terms (unified/fundamental/etc.)
>
> We agree that some of our wording in the initial submission may sound overly strong. Our intention with terms like “unified” was not to claim a universally fundamental principle, but to highlight a concrete technical property: the same debiased energy $E$, instead of different corrections, is used in different parts of the pipeline. To avoid any impression of overclaiming, we will revise the manuscript to replace adjectives such as “unified” or “universal” with more measured terms.

---

> ### Author Response · Authors · 2025-11-27
>
> We sincerely thank you for your detailed and critical feedback, which has helped us substantially refine the positioning and analysis of our method. We have revised the paper and added focused experiments and discussions in Appendices E.1, E.3, and E.5 to better address your concerns, and we would be very thankful if you could briefly revisit the manuscript and our responses to see whether the revisions alleviate your worries before the rebuttal period closes.

---

### Official Review · Reviewer_A5KJ · 2025-10-31

**Soundness:** 3
**Presentation:** 3
**Contribution:** 2
**Rating:** 6
**Confidence:** 4

**Summary:**

This paper makes the following key contributions for long-tailed semi-supervised learning (LTSSL), suitable for international machine learning conference review:

1. **Unified Bias-Aware Objective**: Proposes Bias-Aware Loss (BiAL), which replaces static distribution priors (limiting existing methods) with the model’s current class bias (estimated from no-information inputs). BiAL unifies bias correction across cross-entropy/logit adjustment and contrastive heads, and extends to supervised learning, enabling consistent debiasing in diverse architectures (e.g., FixMatch, CCL).

2. **Theoretical Guarantees**: Establishes Fisher consistency for balanced error rate (BER) with BiAL’s debiased energy, derives dynamic-regret advantages under prior drift (induced by pseudo-labeling), and proves that static-prior methods suffer from unavoidable mismatch—quantifying their excess BER to justify BiAL’s adaptivity.

3. **Plug-and-Play Implementation**: Adds negligible computational overhead (only lightweight bias probing and logit adjustment) without extra components. It integrates seamlessly into existing SSL pipelines via warm-up/ramp scheduling for stability.

4. **Empirical Validation**: Achieves state-of-the-art (SOTA) performance on CIFAR10/100-LT, STL10-LT, and ImageNet-127 across consistent/uniform/reverse unlabeled distributions. It concurrently improves pseudo-label quality and test accuracy, outperforming strong baselines (e.g., CPE, Meta-Experts, CCL).

**Strengths:**

# 1. **Well-motivated and unified formulation**
The paper proposes a unified Bias-Aware Loss (BiAL) that replaces static class priors with model-induced bias estimated from no-information inputs. This principled abstraction enables a plug-and-play correction mechanism compatible with multiple paradigms such as CE, LA, and contrastive heads . The idea is conceptually clean and addresses a central limitation of prior long-tailed SSL approaches.

# 2. **Solid theoretical justification**
The authors provide clear theoretical insights, showing that pseudo-labeling induces dynamic class-prior drift and that static prior correction becomes misspecified. The analysis within the balanced-error framework demonstrates that BiAL can reduce dynamic regret and align more closely with the evolving effective prior This theoretical grounding strongly supports the method.

**Weaknesses:**

# 1. **Bias estimation stability remains unclear**
The bias is estimated using model predictions on no-information images (e.g., black images) and stabilized with EMA and warm-up strategies . However, the accuracy and robustness of such estimation—particularly during early training—may be questionable. More analysis on sensitivity to batch size, input type, and noise would be helpful.
# 2. **Performance improvements can be marginal in some setups**
Although the method achieves state-of-the-art results on several benchmarks, improvements over strong baselines (e.g., FixMatch+ACR/CPE) are sometimes relatively small and appear within statistical variance. More significance analysis or discussion would help contextualize these gains.

**Questions:**

See weakness

---

> ### Author Response · Authors · 2025-11-21
>
> We are sincerely grateful to the reviewer for the careful reading and the very positive overall assessment of our work. These encouraging remarks were very helpful in guiding the revision. And below we address the two main concerns.
>
> Q1 Bias estimation stability remains unclear
>
> We fully agree that the stability of the estimated bias is crucial, especially in the early phase of training when the model has not yet learned meaningful features. This is why our design explicitly introduces a warm-up period and a ramp schedule. As described in the main text and further analyzed in Appendix E.3, the debiasing term is kept inactive during the first $E_{\text{warm}}$ epochs, and the coefficient $\beta_t$ is then increased gradually over an additional $E_{\text{ramp}}$ epochs. In other words, BiAL is deliberately “switched off” while the backbone is still highly unstable, and only starts to influence the optimization once the classifier has acquired basic discriminative ability. The new sensitivity analysis in E.3 shows that varying $E_{\text{warm}}$ and $E_{\text{ramp}}$ over a fairly wide range leads to only minor changes in performance, and all of these configurations remain better than the corresponding base SSL framework. This indicates that BiAL does not rely on finely tuned schedules: it is enough to adopt a conventional warm-up and ramp, as is also done in prior debiasing methods (CDMAD[1], CCL[2]), to avoid the early-noise issue that the reviewer is concerned about.
>
> In addition, we now provide a more systematic study of the no-information probes used to estimate the bias, in Appendix E.5. There we replace the original all-black baseline with a variety of alternatives, including gray, white, colored constants, Gaussian noise, Gaussian-blurred patterns, and even an out-of-range constant “non-image”. Under a fixed training configuration for BiAL-FixMatch and BiAL-CCL on CIFAR-10-LT, all constant-color and blurred baselines yield almost identical accuracy, and the all-black baseline is only slightly better than the others. This evidence supports the intuition that, for convolutional networks with normalization, spatially constant inputs are effectively featureless and differ mainly by a global offset that is largely removed in early layers, so the resulting bias vector reflects the model’s intrinsic class preference rather than the specific RGB values. Together with the warm-up and ramp mechanisms and the EMA used to smooth the estimates, these results suggest that our bias estimation is both structurally robust and practically stable, and not overly sensitive to the precise choice of hyperparameters or probe pattern.
>
>
> Q2 Magnitude and significance of performance improvements
>
> In the current version, we report mean and standard deviation over multiple random seeds in the main tables and appendices. Across datasets, imbalance regimes, BiAL consistently improves over each base SSL method. This indicates that the gains are statistically meaningful rather than artifacts of training noise.
>
> At the same time, it is important to distinguish between different baselines. CCL is an already highly complex pipeline, combining a dual-branch architecture, reliable pseudo-label, smoothed pseudo label, and an energy mask mechanism. In such a setup, there is naturally limited headroom left for further improvement, so the overall gains of BiAL-CCL are moderate, even though BiAL still improves tail performance and balanced error. To make the effect of BiAL more visible, we have added new experiments on a simpler method, ACR[3], in Appendix E.1 (Table 6). BiAL-ACR achieves noticeably larger improvements over the original ACR in overall accuracy. Together with the results already reported in Tables 1 and 2 for BiAL-FixMatch, where BiAL yields clear gains on top of a simpler backbone, these experiments show that when the underlying SSL method leaves more room for correcting dynamic bias, BiAL can provide substantial improvements; when the baseline is already heavily engineered, BiAL behaves more like a stabilizing bias-aware refinement, offering smaller but consistent gains.
>
> [1] Class-Distribution-Mismatch-Aware Debiasing for Class-Imbalanced Semi-Supervised Learning. CVPR 2023
>
> [2] Continuous Contrastive Learning for Long-Tailed Semi-Supervised Recognition. Neurips 2024
>
> [3] Towards Realistic Long-Tailed Semi-Supervised Learning: Consistency Is All You Need. CVPR 2023

---

> ### Author Response · Authors · 2025-11-27
>
> We are very grateful for your positive overall assessment and for the constructive concerns regarding bias stability and the practical significance of the gains. In response, we have expanded the experiments and analyses in Appendices E.1, E.3, and E.5, and we would greatly appreciate it if you could take a quick look at the updated version and let us know whether these clarifications sufficiently address your questions before the rebuttal deadline.

---

### Official Review · Reviewer_ao8U · 2025-10-31

**Soundness:** 2
**Presentation:** 3
**Contribution:** 2
**Rating:** 4
**Confidence:** 4

**Summary:**

This paper introduces Bias-Aware Loss (BiAL), a unified framework for long-tailed semi-supervised learning (LTSSL) that replaces static distribution priors with the dynamically estimated bias of the model. The core idea is to measure this bias by probing the model on no-information inputs (e.g., solid black images) and then use it to correct logits throughout training and inference. The method is simple, theoretically grounded and empirically strong. It achieves highly competitive performance across multiple datasets (CIFAR-10/100-LT, STL-10-LT, ImageNet-127). The paper provides good theoretical guarantees (Fisher consistency, dynamic regret bounds) and several experiments that validate the method's robustness.

**Strengths:**

1. Novel conceptual framework
    - Introduces a dynamic bias estimation mechanism that generalizes prior static approaches.
    - Unifies existing bias-corrective losses under a single principle.
2. Strong theoretical foundation
    - Fisher consistency and dynamic regret proofs provide mathematical backing.
    - Gradient-level analysis clarifies improvements in minority class margins.
3. Comprehensive empirical validation
    - Benchmarks across 4 datasets and multiple distribution regimes.
    - Thorough ablations and sensitivity checks.
4. Practical utility
    - Plug-and-play integration with minimal overhead.
    - Includes practical engineering details (warm-up, EMA smoothing, ramp-up).
    - Clear implementation and reproducibility potential.
5. The paper is well written, with clear motivation, good organization, and visual presentation.

**Weaknesses:**

**Limited bias source analysis:** The method captures aggregate bias, but the paper does not separate its components (e.g. class imbalance vs. architectural difficulty).

**Hyperparameter sensitivity:** The introduction of new parameters $(\beta, E_\mathrm{est}, E_\mathrm{warm}, E_\mathrm{ramp}, \alpha)$ adds to the tuning cost. While a sensitivity analysis is provided, clear heuristics for setting these on new datasets are limited.

**Minor writing issues:** Includes missing references to figures and a minor equation labeling error, which slightly impact the reading flow.

**Limited exploration of "No-Information" inputs:** The paper uses all-black images but does not ablate this choice. Exploring other types of non-informative inputs (e.g., noise patterns) could have strengthened the methodological foundation.

**Domain-specific semantic meaning of "No-Information" inputs:** The method's core assumption is that a solid black image serves as a neutral, non-informative baseline. However, in specialized domains like medical imaging, the color black can carry significant clinical meaning (e.g. specific tissue types, or the absence of a finding). In such cases, using a black image would not probe the model's intrinsic class bias but would instead measure its response to a semantically charged input, leading to a corrupted and misleading bias estimate.

**Contextual bias from training data:** The bias estimation relies on the model's output for a constant-colour input. However, if the original training dataset contains correlations between plain backgrounds and specific classes, the model may learn these spurious associations. Consequently, the estimated bias vector $b_\theta$ would capture this dataset-specific contextual bias (e.g., a bias towards classes frequently appearing with blank slides) alongside the intended class-frequency bias.

**Questions:**

1. Bias composition: The measured bias $b_\theta$ is treated as a unified vector. Can you disentangle how much of this bias originates from class imbalance versus other factors, such as the model's architectural prior or dataset-specific visual biases (e.g., background correlations)? Is the bias on no-information inputs a pure reflection of the label distribution?
2. Hyperparameter tuning: What heuristics or adaptive schemes could help set $\beta, E_\mathrm{warm}$, and $E_\mathrm{ramp}$ for new datasets?
3. Scalability: How does computational cost scale with model size and class count?
4. Failure cases: When might BiAL underperform relative to static priors?
5. Bias estimation robustness: What happens when bias estimation is noisy or unstable?
6. Visual similarity: Could you analyse how BiAL affects discrimination among visually similar head vs. tail classes?
7. Experimental fairness: Were the same data samples used consistently across all method comparisons?

---

> ### Author Response · Authors · 2025-11-21
>
> We thank the reviewer for the detailed and constructive feedback. Regarding the minor writing issues in the weakness section, we have revised the manuscript to fix the missing figure references, which we hope improves the overall readability.
>
> Q1 Bias composition
>
> We agree that the vector $b_\theta$ measured on no-information inputs should be viewed as an aggregate bias signal rather than a “pure” estimator of the label-frequency prior only. Conceptually, BiAL is designed to correct the model with respect to its effective training prior, which naturally mixes several sources: the long-tailed label distribution, the dynamics introduced by pseudo-labeling and architectural biases of the models. Our goal is not to disentangle these components explicitly, but to neutralize the net tendency of the model to over-favor certain classes when there is no class-specific evidence in the input. At the same time, any residual bias due to architecture or background correlations is also part of what we would like to cancel out, because it also manifests discriminative structure.
>
> Moreover, in our experiments on different SSL backbones such as FixMatch[1] and ReMixMatch[2], we find that even on the same dataset the estimated bias can differ substantially across models. This suggests that trying to “debias the dataset only” or designing a correction tailored to a single fixed model is both difficult and of limited practical value: in realistic pipelines, the effective bias always arises from the interaction between data, architecture, and training dynamics. For this reason, we deliberately do not insist on separating “dataset bias” from “model bias”; both manifest as structural discrimination with respect to the input and should be removed at the decision level. Regarding the concern that using a baseline image that resembles typical backgrounds might inadvertently interact with dataset bias, our theory section and the ablations in Appendix E.5 show that what BiAL needs is simply a no-information input, i.e., one that contains no class-discriminative spatial structure. Under standard convolutional backbones with normalization layers, the exact color of such a structure-free input (even a “non-image” constant tensor) has negligible influence on the resulting bias vector; different constant baselines lead to almost identical $b_\theta$ and downstream performance, whereas adding strong noise or artefacts is what actually hurts. This also aligns with human intuition: a plain red background is not perceived as an “apple” by itself, and the model should likewise not infer a specific class solely from an unstructured background. We therefore clarify that $b_\theta$ should be interpreted as an effective prior capturing all these sources together, and that BiAL is explicitly designed to compensate this aggregate tendency rather than any single isolated component.
>
>
> Q2 Hyperparameter tuning
>
> We have added a more detailed sensitivity study in Appendix E.3, where we vary $E_{\text{warm}}$, $E_{\text{ramp}}$ and $\beta$. The empirical results show that BiAL is quite robust to these choices: once the model has passed a basic learning phase, the exact warm-up and ramp lengths have only a mild impact on the final accuracy. The motivation for introducing these three parameters is not to increase tuning burden, but to provide a simple way to control when BiAL starts to act and how strongly it debiases. In practice, we recommend a straightforward heuristic for new datasets: start from $\beta=1.0$, which already performs well in almost all our experiments, choose $E_{\text{warm}}$ so that BiAL is switched on only after the base SSL method has reached a reasonable training loss (for example, a fraction of the total epochs that is similar to what we use on CIFAR10/100-LT and STL10-LT), and set $E_{\text{ramp}}$ to cover the subsequent transition period where the debiasing strength increases smoothly from zero to its target value. Our ablations indicate that any setting where $E_{\text{warm}}$ leaves sufficient time for BiAL to operate, and $E_{\text{ramp}}$ avoids an abrupt jump, tends to give very similar performance, which suggests that these hyperparameters offer stability control rather than a fragile knob that must be finely tuned.
>
> Q3 Scalability
>
> We have included a complexity analysis in Appendix E.4. BiAL does not introduce any extra backbone or classifier; it only requires computing the bias vector $b_\theta$ on no-information inputs and subtracting $\beta_t b_\theta$ from the logits. This leads to a computational cost that scales linearly with the number of classes in exactly the same way as a standard logit post-processing step. On all reported datasets and architectures, the overhead remains within a few percent of the base SSL training time. Thus, BiAL can be regarded as a lightweight plug-in whose cost is negligible compared to the backbone training.
>
> [1] FixMatch. Neurips 2020
>
> [2] ReMixMatch. ICLR 2020

---

> ### Author Response · Authors · 2025-11-21
>
> Q4 Failure cases
>
> We agree that BiAL can underperform relative to static priors in certain regimes. The most prominent case is the very early stage of training, when the classifier has not yet learned a meaningful decision boundary and the estimated bias $b_\theta$ is highly unstable. Because BiAL is intentionally data-driven and adapts to the evolving model state, applying it from the first epoch can overreact to noisy predictions and temporarily push the model away from a good solution, whereas a static prior remains unchanged and therefore appears more stable in that particular phase. This is precisely why we introduce the warm-up and ramp epochs $E_{\text{warm}}$ and $E_{\text{ramp}}$: BiAL is kept inactive until the base SSL model has acquired basic discrimination ability, and its strength is then increased gradually rather than in one abrupt step. In experiments, we indeed observe that without warm-up BiAL may hurt performance on some datasets, but with the warm-up and ramp schedule the early instability is greatly alleviated and BiAL consistently outperforms static priors over the full training run.
>
> Q5 Bias estimation robustness
>
> When the bias estimate itself is noisy or unstable, BiAL can have a negative effect; this happens in two main situations. The first is again the early training regime, where the classifier is under-trained and its predictions on any input fluctuate strongly. The second is when the “baseline” inputs used to compute $b_\theta$ are themselves contaminated by noise or artefacts, as in the ablations in Appendix E.5 where we intentionally use heavily noisy or blurred images. In these settings, the resulting $b_\theta$ is no longer a clean summary of the model’s global preference and may inject additional noise into the logits, which can degrade performance. Our design choices are aimed at mitigating exactly these effects. By delaying BiAL with $E_{\text{warm}}$, we avoid relying on bias estimates in the most unstable phase. By using no-information inputs that contain no semantic structure (constant-colour images) and averaging the outputs over a small batch, we obtain a smooth, low-variance estimate of $b_\theta$. In Appendix E.5 we show that different structure-free baselines yield very similar bias vectors and downstream accuracy, whereas adding strong noise or blur is what actually harms robustness. In practice, combining no-information images with the warm-up and ramp schedules makes bias estimation stable enough that BiAL behaves like a gentle, global correction rather than a noisy perturbation.
>
> Q6 Visual similarity
>
> Conceptually, BiAL operates by subtracting a global bias vector $b_\theta$ from all logits, and this vector is estimated only from no-information inputs. As a result, BiAL does not inject any new sample-dependent pattern into the representation space and does not directly modify the similarity relations between individual images or between visually similar classes. What it does is to remove the global advantage that head classes enjoy. For a head–tail pair of visually similar classes, a biased model tends to predict the head label whenever the features are ambiguous, because the head logit is lifted by the effective prior while the tail logit is suppressed. Subtracting $b_\theta$ shifts the logits of all classes by a fixed amount that reflects this global preference. In ambiguous regions, the gap between the head and tail logits shrinks, so the decision is driven more by the actual visual evidence than by imbalance-induced bias.
>
> Importantly, because the baseline input used to compute $b_\theta$ is a structure-free, no-information image, it does not resemble any particular object or background configuration in the dataset. Under standard convolutional architectures with normalization, different choices of such no-information baselines yield almost identical $b_\theta$ and do not create new confusion between semantically or visually similar categories. This behaviour is confirmed in our ablations in Appendix E.5, where we observe that all structure-free baselines lead to very similar bias vectors and class-wise performance, and that degradation only appears when the baseline is corrupted with strong noise or artefacts. Thus, BiAL does not increase ambiguity between similar classes; rather, by cancelling the global head bias estimated from no-information inputs, it allows those classes to compete on a more balanced footing based on their true visual similarity. This is consistent with our gradient-level analysis, which shows that BiAL enlarges minority margins without requiring class-specific reweighting.
>
> Q7 Experimental fairness
>
> For experimental fairness, all methods in our comparisons are trained and evaluated under exactly the same protocol.

---

> ### Author Response · Authors · 2025-11-27
>
> Thank you again for your careful review and for raising several important questions about the bias composition, hyperparameters, and the choice of no-information inputs. We have added targeted analyses in Appendices E.1, E.3, and E.5 to address these points, and would highly appreciate it if you could briefly revisit our responses and the revised appendices in case you have further suggestions before the rebuttal period ends.

---

> > ### Comment · Reviewer_ao8U · 2025-11-27
> > **Follow-up on Q1 response**
> >
> > 1. While the explanation that BiAL aims to correct an aggregate bias is appreciated, the original concern remains. Although the paper repeatedly relies on the claim that $b_\theta$ “monotonically reflects” the effective training prior (1)
> > (Section 3.5), there is no empirical evidence demonstrating that this correspondence actually holds in practice. The paper does not provide controlled experiments disentangling how much of $b_\theta$ comes from label imbalance versus architectural or pseudo-label–induced biases, nor does it verify that changes in the effective prior lead to predictable changes in the estimated bias vector.
> >
> > (1) $\mathrm{log} \tilde{\pi}_t^{PL}$

---

> > > ### Comment · Reviewer_ao8U · 2025-11-28
> > > **follow-up on Q2-Q7 responses**
> > >
> > > 2. The additional sensitivity analysis and the qualitative guidance on setting $\beta, E_{warm}, \text{and} E_{ramp}$ largely address the concern. It may still be helpful to provide a slightly more concrete or operational heuristic for new datasets. However, overall, the issue is sufficiently resolved.
> > >
> > > 3. The authors have now provided a complexity analysis in Appendix E.4, clarifying that BiAL does not introduce additional backbones and adds only a logit-level correction whose cost scales linearly with the number of classes, similar to standard post-processing. The extra forward passes on no-information inputs incur only a small overhead (a few percent of SSL training time in their setups). This addresses the concern about scalability with respect to both model size and class count.
> > >
> > > 4. Beyond early training instability, concerns remain about the interpretation of the bias vector in domains where constant-color inputs are not semantically neutral. The current method estimates $b_\theta$ from the model’s response to a constant image, but if the training data exhibits correlations between near-uniform backgrounds and specific classes (e.g., blank slides or uniform regions associated with particular labels in medical imaging), then this “no-information” baseline is not actually context-free. In such cases, the estimated bias vector $b_\theta$ may conflate class-frequency imbalance with dataset-specific contextual bias, capturing a preference for classes that frequently co-occur with plain backgrounds rather than an intrinsic prior. The claim that constant baselines are structure-free and domain-neutral seems well supported for the image datasets examined, but its validity in more specialized domains remains unclear.
> > >
> > > 5. The authors’ analysis and ablations largely address the concern about the standard SSL benchmarks, and the combination of warm-up, ramp, and constant-color baselines appears reasonably robust in those settings. However, while the provided analysis addresses robustness for standard SSL datasets, additional consideration or validation would be required for domains where constant-color images are not semantically neutral. This aspect remains somewhat problematic in terms of broader applicability.
> > >
> > > 6. The authors’ explanation clarifies how BiAL reduces global head-class bias without distorting local similarity structure, and the provided ablations support this for the natural-image SSL datasets studied.
> > >
> > > 7. Thanks for the explanation.
> > >
> > > Overall, the authors have responded thoroughly to the questions and clarified several technical aspects of the method. Many of the practical concerns, such as hyperparameter sensitivity, scalability, failure cases, such as during early training, and the impact on visually similar classes, have been addressed. The rebuttal provides additional empirical evidence and clearer motivation for the design choices. However, the central issue regarding the nature and interpretation of the bias vector remains only partially resolved. It is still unclear to what extent the estimated bias reflects label imbalance versus dataset or architecture-specific artifacts, and the reliance on constant-color “no-information” inputs raises concerns for domains where such inputs are not semantically neutral. In such settings, the estimated bias can itself become a biased bias, shaped by contextual or domain-specific factors rather than representing a clean global prior. While the rebuttal improves the understanding of the method and addresses several secondary points, this core limitation persists and constrains the generalization of the approach.

---

> > > > ### Author Response · Authors · 2025-12-01
> > > >
> > > > Thank you again for the very careful follow-up and for acknowledging that most of the practical concerns (hyperparameter sensitivity, scalability, early-training stability, visually similar classes, etc.) have been addressed. We fully agree that the remaining issue is how to interpret the estimated bias vector and how broadly this interpretation can be generalized beyond the natural-image SSL benchmarks considered in our experiments.
> > > >
> > > > Our intention in the paper is to use the estimated bias as an aggregate, task-effective summary of the model’s global preference over classes, rather than as a perfectly disentangled estimate of a “pure” label prior. By construction, it inevitably mixes the effects of class imbalance, architecture and pseudo-label dynamics, and BiAL aims to counteract the resulting systematic head-class preference in this aggregate sense. In the revised version, we will explicitly soften the wording around “reflecting the effective prior” and clearly state that we do not claim to fully separate different sources of bias; instead, we focus on whether this aggregate quantity is useful for debiasing long-tailed SSL training.
> > > >
> > > > We also take the concern about domains where constant-color “no-information” inputs are not semantically neutral very seriously. In this submission we deliberately restricted our empirical scope to standard natural-image LTSSL datasets, where our ablations suggest that constant baselines behave as intended. In future versions, we plan to investigate more specialized domains (such as medical or scientific imaging) and to validate BiAL with alternative, domain-appropriate low-information probes (e.g., noise patterns or masked regions) that avoid encoding dataset-specific context. We will explicitly discuss this as a key limitation and direction for further study in the revised manuscript. We very much appreciate the reviewer for highlighting this point so clearly, and we hope these clarifications make the scope and limitations of our approach more transparent.

---

### Official Review · Reviewer_XgAb · 2025-11-02

**Soundness:** 2
**Presentation:** 2
**Contribution:** 2
**Rating:** 4
**Confidence:** 4

**Summary:**

This paper studies the problem of long-tailed semi-supervised learning (LTSSL), where both label imbalance and pseudo-label noise cause strong class bias during training. The authors observe that most existing debiasing methods use static distribution priors (e.g., class frequencies), which become inaccurate as the model evolves and pseudo-labels change the effective class distribution.

To address this, the paper proposes Bias-Aware Loss (BiAL), a unified bias-aware objective that replaces static priors with the model’s current bias, estimated directly from its responses to no-information inputs (e.g., blank images). This approach allows consistent correction during both training and inference, and can be easily plugged into different SSL frameworks such as FixMatch and CCL.

**Strengths:**

1. Paper is well-written and overall well organized.

**Weaknesses:**

The main idea of using model bias estimated from no-information inputs is conceptually close to DebiasPL (“Debiased Learning from Naturally Imbalanced Pseudo-Labels,” CVPR 2022). Both approaches rely on the model’s self-bias for correction without external priors. DebiasPL's causal inference pipeline The new method mainly wraps this idea into a unified loss formulation (BiAL) but does not introduce a very different underlying mechanism. The contribution seems more incremental than fundamentally new.

Most experiments are conducted on small benchmarks such as CIFAR10/100-LT and STL10-LT, with limited data diversity and visual complexity. While the method shows nice improvements there, it is unclear whether the gains can generalize to large-scale or real-world long-tailed semi-supervised scenarios (e.g., ImageNet-LT, WebVision, or domain-shifted data).

The paper only discusses classification tasks. It is not clear whether the proposed bias-aware correction can be generalized to other settings like detection, segmentation, or multimodal learning. Since those tasks often involve structured outputs and continuous predictions, the practical applicability of BiAL outside classification remains uncertain.

Marginal performance gains. As shown in table 1 and table 2, the performance gains is often within 0.5%.

**Questions:**

Please check the weakness section.

---

> ### Author Response · Authors · 2025-11-21
>
> We sincerely thank you for the careful reading of our paper and for the positive feedback on the writing and organization. And below we address the four main concerns.
>
> Q1 more incremental than fundamentally new
>
> Regarding the relation to DebiasPL[1], we agree that there is a conceptual similarity in that both approaches leverage the model’s own behaviour, rather than an externally specified prior, to perform debiasing. However, the way bias is defined, estimated, and integrated into training is quite different. DebiasPL is built around the pseudo-label distribution on unlabeled data: it maintains a momentum-updated class probability vector over unlabeled samples and uses this $\hat p$ to adjust logits (such as via a $-\lambda \log \hat p$ term) and margins in the pseudo-label branch. Its core object is the pseudo-label histogram, and its corrections are applied primarily to the unlabeled branch of specific pipelines such as FixMatch. In contrast, BiAL defines an effective training prior $b_\theta$ by probing the model with no-information inputs and then constructs a debiased energy $E(x) = z(x) - \beta_t b_\theta$ that serves as a common interface for all loss components in long-tailed semi-supervised learning, including supervised cross-entropy, logit-adjusted heads, and contrastive heads. In this sense, BiAL is not just manipulating pseudo-label statistics but replaces static priors in the underlying objective for both labeled and unlabeled data, and for all heads that consume logits. We also provide a Fisher-consistency and dynamic-regret analysis in the balanced-error-rate setting to justify why adapting to the evolving effective prior is necessary when pseudo-labels continually shift the class distribution, a regime that DebiasPL does not explicitly target. We clarify in Sec F.2 that while BiAL shares the spirit of using model-induced bias, it adopts a different and effective prior estimated from no-information inputs, which is not just a rephrasing of DebiasPL.
>
> Q2 Most experiments are conducted on small benchmarks \& Q3 only discusses classification tasks
>
> We have also included ImageNet-127 in the current version to demonstrate that BiAL extends to a substantially larger and more realistic long-tailed benchmark. And we have restricted both our method and our comparisons to classification tasks: all baselines we discuss (CCL[2], CPE[3], CDMAD[4], ACR[5], Meta-Experts[6], etc.) are designed for and evaluated in long-tailed semi-supervised classification. BiAL is proposed in this same context. While its mechanism is conceptually compatible with heads in detection or segmentation, a careful extension to those settings would require new architectures, datasets, and evaluation protocols that are beyond the space and focus of this work.
>
> Q4 Marginal performance gains
>
> For magnitude of performance gains, we agree that CCL[2] results show relatively modest improvements after adding BiAL. We believe this is largely due to the strength and complexity of CCL itself: it already combines a dual-branch architecture, smoothed pseudo-labels, reliable pseudo-labels, and an energy-based scoring module, which leaves limited headroom for additional components. However, the effect of BiAL is clearer on simpler baselines. In Tables 1 and 2, BiAL+FixMatch provides noticeable improvements, showing that in a standard single-branch SSL pipeline the bias-aware objective brings substantial benefits. In the revision, we have further added experiments with ACR+BiAL, where ACR[5] is a simpler long-tailed SSL baseline than CCL. In these settings, BiAL consistently yields 1–2\% absolute gains, which are comparable to or larger than improvements typically reported for new debiasing modules in LTSSL.
>
> [1] Debiased Learning From Naturally Imbalanced Pseudo-Labels. CVPR 2022
>
> [2] Continuous Contrastive Learning for Long-Tailed Semi-Supervised Recognition. Neurips 2024
>
> [3] Complementary Experts for Long-Tailed Semi-supervised Learning. AAAI 2024
>
> [4] Class-Distribution-Mismatch-Aware Debiasing for Class-Imbalanced Semi-Supervised Learning. CVPR 2023
>
> [5] Towards Realistic Long-Tailed Semi-Supervised Learning: Consistency Is All You Need. CVPR 2023
>
> [6] A Square Peg in a Square Hole: Meta-Expert for Long-Tailed Semi-Supervised Learning. ICML 2025

---

> ### Author Response · Authors · 2025-11-27
>
> We sincerely appreciate your detailed comments and the thoughtful comparison with prior debiasing methods. We have revised the manuscript accordingly, including new analyses in Appendices E.1, E.3, and E.5, and we would be very grateful if you could take a brief look at the updated version and let us know if there are remaining concerns before the rebuttal deadline.

---

### Author Response · Authors · 2025-12-01
**Summary of the discussion for the Area Chair**

Dear Area Chair,

We would like to sincerely thank you and all reviewers for the careful evaluation of our submission. Our work proposes BiAL, a bias-aware energy formulation for long-tailed semi-supervised learning that replaces logits with a simple debiased energy shared by supervised, SSL, and contrastive heads and by test-time prediction, together with a theoretical analysis under dynamic class-prior drift.

During the rebuttal, we made several focused revisions to address the main concerns. In App.E.1 we added new experiments on ACR+BiAL, showing that BiAL consistently improves a simpler LA-based baseline beyond CCL, with clearer gains on CIFAR10/100-LT. In App.E.3 we conducted a more detailed sensitivity study of the warm-up and ramp schedules, indicating that BiAL is robust to the choices of $E_\text{warm}$ and $E_\text{ramp}$ and does not require delicate tuning. In App.E.5 we systematically ablated the choice of “no-information” probes, and found that all constant/low-frequency baselines behave similarly while high-variance noise degrades performance, supporting the design used on natural-image LTSSL benchmarks.

For the Reviewer XgAb.
Reviewer XgAb mainly questioned the novelty of BiAL relative to DebiasPL and the scope of the experiments. In our response and revision, we clarified the difference between post-hoc or stage-wise corrections and our loss-level unified energy that is consistently used by all heads and for inference, and we pointed to the dynamic regret analysis in Sec.3.5 and App.B. We also added the ACR+BiAL results in App. E.1 to demonstrate that the method brings gains beyond CCL, and emphasized the results on ImageNet-127 as a large-scale benchmark. Remaining disagreement is mostly about how incremental the contribution is judged to be, rather than about correctness or experimental soundness.

For the Reviewer ao8U.
Reviewer ao8U provided a very detailed and constructive review, focusing on bias stability, hyperparameters, complexity, failure cases, and especially the interpretation of the bias vector and the role of constant-color probes. The new analyses in App.E.3 and E.5, together with the existing complexity study, largely resolve the concerns on sensitivity, scalability, and early training behavior; the reviewer explicitly notes that these issues are “sufficiently resolved” on the natural-image SSL datasets. The reviewer still highlights an important limitation: in domains where constant-color inputs are not semantically neutral (certain medical or scientific imaging setups), the estimated bias may mix global imbalance with dataset-specific background correlations. In our follow-up we acknowledged this as a limitation of the current instantiation, clarified that we use the bias as an aggregate task-effective proxy rather than a pure prior, and committed to make this limitation explicit in the paper and to discuss future extensions using domain-appropriate low-information probes.

For the Reviewer A5KJ.
Reviewer A5KJ gave the most positive score and praised the motivation and unified formulation, but asked for more evidence on bias stability and on the practical significance of the performance gains. In response, we highlighted the warm-up and ramp design and added the extended sensitivity study in App.E.3, showing that BiAL is robust to a wide range of $E_\text{warm}$ and $E_\text{ramp}$ settings. We also report mean $\pm$ std over multiple seeds, and we added the ACR+BiAL experiments in App.E.1 to complement the clearer gains already present on FixMatch in Tables 1-2. The remaining comments mainly concern how to interpret ``small" gains on top of very strong baselines like CCL; we clarified that BiAL brings larger improvements on simpler baselines and on tail classes, while acting as a stabilizing refinement when the base method is already highly engineered.

For the Reviewer waeb.
Reviewer waeb raised the sharpest comparison with CDMAD/LA and questioned whether BiAL is essentially another logit adjustment. We clarified in the main text and App.F.1 that, while conceptually related, BiAL differs in mechanism (a single debiased energy used in all losses and at test time, instead of localized post-hoc corrections) and in analysis (explicit dynamic BER/regret under prior drift). Motivated by this comment, we also performed additional experiments showing that naively stacking CDMAD on top of CCL severely degrades performance, whereas BiAL-CCL and BiAL-ACR remain stable and often improve over their baselines, indicating that the loss-level formulation matters in practice.

Overall, we believe the revised submission now presents a clearer picture of what BiAL contributes: a practically simple yet coherent bias-aware energy that can be plugged into existing LTSSL pipelines, supported by both theory and extensive experiments, while being explicit about its limitations and domain of applicability. We are very grateful for your time and consideration in making the final decision.

---

### Meta-Review · Area_Chair_QuWG · 2026-01-06

**Summary:**

The paper deals with the problem of long-tail distributions in semi-supervised learning and proposes a Bias-Aware Loss
(BiAL) based on models response to no-information input obtained by forwarding black images.
the evaluation is done on CIFAR10LT, CIFAR100-LT, STL-10, and ImageNet-127 datasets and the method shows a consistent marginal improvment compared to the baseline of FixMatch + CCL.

**Reviewer Concerns:**

Please see the summary of concerns under Reviewer Scores.

**Reviewer Scores:**

The paper was considered by four reviewers.
Reviewer XgAb had an inital rating of 4 (confidence 4). Main concerns were 1) limited novelty compared to other methods (DebiasPL). 2) experiments only on small-scale data, 3) limitations to classification, and 4) marginal improvements.
Authors addressed 1) in Sec F.2 elaborating the difference between DebiasPL and Bail, objections 2) and 3) are more or less rejected and 4) is addressed by showing more evaluation e.g. ARC+Bial and BiAL-LDAM-DRW.

While ARC+Bial and BiAL-LDAM-DRW shows a slight improvement, I would consider this the only points that might address the question. I'm not sure if this would be the experiment the reviewer was looking for to address the overall issues, also with respect to concerns 1)-3). I would in this case assume that the reviewer might have kept the score in an after rebuttal discussion.

Reviewer ao8U had an inital rating of 4 (confidence 4). Main concerns were 1) Limited bias source analysis, 2) Hyperparameter sensitivity, 3) Minor writing issues, 4) Limited exploration of "No-Information" inputs, 5) Domain-specific semantic meaning of "No-Information" inputs, and 6) Contextual bias from training data.

The last comment from this reviewer was: "Overall, the authors have responded thoroughly to the questions and clarified several technical aspects of the method. Many of the practical concerns, such as hyperparameter sensitivity, scalability, failure cases, such as during early training, and the impact on visually similar classes, have been addressed. The rebuttal provides additional empirical evidence and clearer motivation for the design choices. However, the central issue regarding the nature and interpretation of the bias vector remains only partially resolved. It is still unclear to what extent the estimated bias reflects label imbalance versus dataset or architecture-specific artifacts, and the reliance on constant-color “no-information” inputs raises concerns for domains where such inputs are not semantically neutral. In such settings, the estimated bias can itself become a biased bias, shaped by contextual or domain-specific factors rather than representing a clean global prior. While the rebuttal improves the understanding of the method and addresses several secondary points, this core limitation persists and constrains the generalization of the approach."

I would interpret this as a statement that the reviewer did not consider raising his/her score, and therefore consider a score of 4.

Reviewer A5KJ had an inital rating of 6 (confidence 4). The main concerns were that 1) bias estimation stability remains unclear and 2) that performance improvements are marginal.

While both of those comments were addressed (see also other reviews), I don't think that the additional information would have pushed the reviewer to change his score of 6 to lower or higher. I would therefore assume a final score of 6 here as well.

Reviewer waeb had an inital rating of 2 (confidence 4). The main concerns were 1) the limited novelty ( that using the model's response to no-information inputs to estimate and correct for its bias has been proposed in CDMAD), 2) the lack theoretical foundationand 3) the qulaity of writing (The paper is replete with grandiose terms like unified, universal, and "fundamental," which do not align with the actual substance of the contribution).

The authors mainly address points 1) and 2), and fixed 3). Mainly based on the extensive answer to 1) I could imagine this reviewer changing his stance and increasing the score by two points to 4. I would therefore assume a final rating of 4 in this case.

Overall, after reviewing the discussion, my best guess is that the final scores for the paper could have been 4, 4, 6, and 4, resulting in a final score of 4.5 and thus falling below the acceptance threshold.

While I appreciate the idea of the paper, I understand the feedback of the reviewers, especially with respect to small training regimes and limited evaluation, which could have only been partially resolved during the rebuttal. I would therefore recommend rejecting the paper and encouraging the authors to submit a revised version to a follow-up venue.

---

### Decision · Program_Chairs · 2026-01-26

Reject